# ALMOST SURE CONVERGENCE OF STOCHASTIC HAMILTONIAN DESCENT METHODS

## ABSTRACT

Gradient normalization and soft clipping are two popular techniques for tackling instability issues and improving convergence of stochastic gradient descent (SGD) with momentum. In this article, we study these types of methods through the lens of dissipative Hamiltonian systems. Gradient normalization and certain types of soft clipping algorithms can be seen as (stochastic) implicit-explicit Euler discretizations of dissipative Hamiltonian systems, where the kinetic energy function determines the type of clipping that is applied. We make use of dynamical systems theory to show in a unified way that all of these schemes converge to stationary points of the objective function, almost surely, in several different settings: a) for $L-$smooth objective functions, when the variance of the stochastic gradients is possibly infinite b) under the $(L_0, L_1)-$smoothness assumption, for heavy-tailed noise with bounded variance and c) for $(L_0, L_1)-$smooth functions in the empirical risk minimization setting, when the variance is possibly infinite but the expectation is finite.

## 1 INTRODUCTION

In this article we consider the optimization problem

$$\min_{q \in \mathbb{R}^d} F(q), \tag{1}$$

where $F : \mathbb{R}^d \to \mathbb{R}$ is an objective function. A common case in mathematical statistics and machine learning is the empirical risk minimization setting, where $F$ is a weighted sum of loss functions:

$$F(q) = \frac{1}{N} \sum_{i=1}^{N} f_i(q), \tag{2}$$

with $f_i(q) = \ell(h(x_i, q), y_i)$. Here, $\{(x_i, y_i)\}_{i=1}^{N}$ is an underlying data set of feature-label pairs in the feature-label space $\mathcal{X} \times \mathcal{Y}$, $h(q, \cdot)$ is a model with model parameters $q$ such as a neural network or a regression function, and $\ell$ is a loss function. A common approach within the machine learning community for solving problems of the type given by (1) is to employ *stochastic gradient descent* (SGD) (Robbins & Monro, 1951). The solution to (1) is approximated iteratively with a stochastic approximation to the gradient of the function defined by (2):

$$q_{k+1} = q_k - \alpha_k \nabla f(q_k, \xi_k). \tag{3}$$

Here $\alpha_k$ is the learning rate and $\xi_k$ is a random variable that accounts for the stochasticity. A common choice is to take a random subset $B_{\xi_k} \subset \{1, \dots, N\}$ of the indices of the objective function defined by (2) and choose

$$\nabla f(q, \xi_k) = \frac{1}{|B_{\xi_k}|} \sum_{i \in B_{\xi_k}} \nabla f_i(q), \tag{4}$$

where $|B_{\xi_k}|$ denotes the cardinality of $B_{\xi_k}$. This is attractive when $N$ is very large and $|B_\xi| \ll N$, as it is less computationally expensive than gradient descent. It also tends to escape local saddle points (Fang et al., 2019) - an appealing property as many machine learning problems are non-convex. Among the variations of SGD is the popular *SGD with momentum*. Its deterministic counterpart

was first introduced in the seminal work of Polyak (1964). A common form of this algorithm is expressed as an update in two stages

$$p_{k+1} = \beta_k p_k - \alpha_k \nabla f(q_k, \xi_k)$$
$$q_{k+1} = q_k + \alpha_k p_{k+1} \tag{5}$$

where $p_0 = 0$ and $\beta_k > 0$ is a momentum parameter. The usage of the momentum update makes the algorithm less sensitive to noise. Indeed, by an iterative argument, we obtain that $p_{k+1} = -\sum_{i=0}^{k} \left( \prod_{j=i+1}^{k} \beta_j \right) \alpha_i \nabla f(q_i, \xi_k)$. That is, $p_{k+1}$ is an average of the previous gradients where $\beta_k$ determines how much we value information from the preceding stages.

Notwithstanding the benefits of stochastic gradient algorithms, they frequently suffer from instability problems such as exploding gradients (Pascanu et al., 2013; Bengio et al., 1994) and sensitivity to the choice of learning rate (Owens & Filkin, 1989). A way to mitigate these issues is to employ gradient clipping (Goodfellow et al., 2016; Pascanu et al., 2012) or gradient normalization. Gradient normalization was introduced in Poljak (1967) in the deterministic and a stochastic version appears already in Andradóttir (1990). A normalized version of the algorithm determined by (3) is given by

$$q_{k+1} = q_k - \alpha_k \frac{\nabla f(q_k, \xi_k)}{\|\nabla f(q_k, \xi_k)\|_2}.$$

In practice a small number $\epsilon > 0$ is added in the denominator to ensure that the update does not become infinitely large.

Gradient clipping was first introduced in Mikolov (2013). In so-called *hard clipping*, the gradient is simply rescaled if it is larger than some predetermined threshold. *Soft clipping*, on the other hand, makes use of a differentiable function for rescaling the gradient (Zhang et al., 2020a). It was recently shown that hard clipping algorithms suffer from an *unavoidable bias term* (Koloskova et al., 2023); a term in the convergence bound that does not decrease as the number of iterations increases. This is one reason why soft clipping is preferable.

## 1.1 Gradient normalization, momentum and Hamiltonian systems

In this article, we study gradient normalization and soft clipping of stochastic momentum algorithms from the perspective of *Hamiltonian systems*. As a first step, we note that if we take $\beta_k = 1 - \gamma \alpha_k$ with $\gamma > 0$, we can view the scheme given by (5) as an approximate implicit-explicit Euler discretization of the equation system

$$\dot{p} = -\nabla F(q) - \gamma p,$$
$$\dot{q} = p. \tag{6}$$

The system 6 is *nearly Hamiltonian* (Glendinning, 1994); taking

$$H(p, q) = F(q) + \varphi(p), \tag{7}$$

with $\varphi(p) = \frac{1}{2}\|p\|_2^2$, we can write it on the form

$$\dot{p} = -\nabla_q H(p, q) - \nabla_{\dot{q}} \mathcal{R}(\dot{q}),$$
$$\dot{q} = \nabla_p H(p, q), \tag{8}$$

where $\nabla_p, \nabla_q$ denote the gradients with respect to $p$ and $q$ respectively and $\mathcal{R}(\dot{q}) = \gamma \frac{\|\dot{q}\|_2^2}{2}$ is a *Rayleigh dissipation function* that accounts for energy dissipation (viscous friction) of the system. Note that this choice of $\mathcal{R}$ yields $\nabla_{\dot{q}} \mathcal{R}(\dot{q}) = \gamma \nabla_p H(p, q)$, which will always be the case in this paper. Thus, for a Hamiltonian of the form 7, (8), reads

$$\dot{p} = -\nabla F(q) - \gamma \nabla \varphi(p),$$
$$\dot{q} = \nabla \varphi(p).$$

We notice that any *fixed point* of this system is a stationary point of $F$, since $(\dot{q}, \dot{p}) = 0$ implies that $\nabla F(q) = 0$. The dissipation term is often included as an extra term in the Euler-Lagrange equations

$$\nabla_{\dot{q}} L(q, \dot{q}) - \frac{\mathrm{d}}{\mathrm{d}t} L(q, \dot{q}) = \nabla_{\dot{q}} \mathcal{R}(\dot{q}),$$

where $L(q, \dot{q}) = \varphi^*(\dot{q}) - F(q)$ is the *Lagrangian*, and $\varphi^*$ is the convex conjugate of $\varphi$, compare Proposition 51.2 and Ex. 51.3 in Zeidler (1985). The physical interpretation is that $q$ is the position of a particle in a potential field $F(q)$ with kinetic energy given by $\varphi^*(\dot{q})$ (in the case when $\varphi(p) = \frac{\|p\|_2^2}{2}$ we have $\varphi^*(\dot{q}) = \frac{\|\dot{q}\|_2^2}{2}$). In many scenarios, such as in this case, it happens that the friction tern is proportional to the velocity (Goldstein et al., 2014). A ball rolling on a rough incline (Wolf et al., 1998; Bideau et al., 1994)) or on a tilted plane coated with a viscous fluid (Bico et al., 2009) could for instance be modelled in this fashion, giving weight to the analogy of the *heavy ball* (Polyak, 1964). See also Goodfellow et al. (2016), for a further discussion on this.

In this paper, we consider generalizations of the algorithm defined by (5) to equations of the type (8) where $F$ is an $L$-smooth, coercive function and $\varphi$ is a convex, coercive and $L$-smooth function. The scheme we consider is given by

$$p_{k+1} = p_k - \alpha_k \nabla f(q_k, \xi_k) - \alpha_k \gamma \nabla \varphi(p_k),$$
$$q_{k+1} = q_k + \alpha_k \nabla \varphi(p_{k+1}), \tag{9}$$

where $p_0 = 0$, $q_0$ is arbitrary, and $\{\xi_k\}_{k \geq 0}$ is a sequence of independent, identically distributed random variables. We show that this scheme converges almost surely to the set of stationary points of $F$. If we take $\varphi(x) = \frac{\|x\|_2^2}{2}$ in (9), we get (5). Taking $\varphi(x) = \sqrt{\|x\|_2^2 + \epsilon}$, $\epsilon > 0$, gives us a gradient normalization scheme, where both the gradient and the momentum variables are rescaled:

$$p_{k+1} = p_k - \alpha_k \nabla f(q_k, \xi_k) - \alpha_k \gamma \frac{p_k}{\sqrt{\|p_k\|_2^2 + \epsilon}},$$
$$q_{k+1} = q_k + \alpha_k \frac{p_{k+1}}{\sqrt{\|p_{k+1}\|_2^2 + \epsilon}}. \tag{10}$$

Other conceivable choices are

    i) *Relativistic kinetic energy:* $\varphi(x) = c\sqrt{\|x\|_2^2 + (mc)^2}$. (Franca et al., 2020)

    ii) *Non-relativistic kinetic energy:* $\varphi(x) = \frac{1}{2}\langle x, Ax\rangle + \langle b, x\rangle + c$, where $A$ is a positive definite, symmetric matrix, $b \in \mathbb{R}^d$ and $c \in \mathbb{R}$. (Goldstein et al., 2014)

    iii) *Gradient rescaling:* $\varphi(x) = c\sqrt{\|x\|_2^2 + \epsilon}$, for $c, \epsilon > 0$.

    iv) *Soft clipping:* $\varphi(x) = \sqrt{1 + \|x\|_2^2}$.

    v) *The symmetric LogSumExp-function:* $\varphi(x) = \ln\left(\sum_{i=1}^d e^{x_i} + e^{-x_i}\right)$, which can be seen as an approximation of the $\ell^\infty$-norm (Sherman, 2013).

    vi) *Half-squared $\ell^p$-norm:* $\varphi(x) = \frac{1}{2}\|x\|_p^2$, for $p \in [2, \infty)$.

Examples i), iii) and iv) are analytically similar, but give rise to different behaviours in the algorithm given by (9). We refer the reader to Beck (2017); Peressini et al. (1993), for verifying that the functions above satisfy the assumptions in Section 5.2.

## 2    CONTRIBUTIONS

Making use of Hamiltonian dynamics, we consider a large class of stochastic optimization algorithms (9) for large-scale optimization problems, for which we perform a rigorous convergence analysis. Our assumptions on the dissipation term $\varphi$ are fairly permissive, and thus the class of algorithms covers both interesting cases like normalized SGD with momentum and various soft-clipping methods with momentum, as well as novel methods. Our analysis shows that the iterates generated by any method in this class are finite almost surely, and that they converge almost surely to the set of stationary points of the objective function $F$. This means that the methods "always" work in practice, in contrast to what can be guaranteed by analyses that show convergence in expectation. These results are valid in many applications, due to fairly weak assumptions on the optimization problem. The exact assumptions are listed in Section 5 but essentially consist of either

    • $L$-smooth objective functions and stochastic gradients with possibly infinite variance, or

- $(L_0, L_1)$-smooth objective functions and heavy-tailed stochastic gradients with bounded variance, or
- $(L_0, L_1)$-smooth objective functions arising in the empirical risk minimization setting and stochastic gradients with possibly infinite variance but bounded expectation.

In particular, we do not assume convexity of the objective function $F$ in any of the cases.

## 3 OUTLINE

In Section 4, we briefly discuss some results that are related to the analysis in this paper. The main results and analysis is presented in Section 5, with conclusions in Section 6. The details of the analysis can be found in Appendix A. This depends on some auxiliary results listed in Appendix B. Finally, Appendix C presents a few numerical experiments that illustrate the behaviour of the methods.

## 4 RELATED WORKS

In the first subsection we consider other formulations of SGD with momentum and how the formulation in this paper relates to them. In the second subsection we summarize work in optimization and statistics which make use of Hamiltonian dynamics. Next, we discuss the approach we use for showing almost sure convergence of the methods. Finally, we discuss the central $(L_0, L_1)-$smoothness condition on the objective function.

### 4.1 MOMENTUM ALGORITHMS

The implementations of SGD with momentum in the libraries Tensorflow (Abadi et al., 2015) and Pytorch (Paszke et al., 2019) are equivalent to (5) after a transformation of the learning rate:

$$p_{k+1} = \beta_k p_k - \alpha_k \nabla f(q_k, \xi_k)$$
$$q_{k+1} = q_k + p_{k+1}.$$

Typically the momentum parameter $\beta_k$ is a fixed number. The update (5) resembles the (hard-clipped) scheme proposed in Mai & Johansson (2021):

$$p_{k+1} = \text{clip}_r \left( (1 - \beta_k) p_k - \beta_k \nabla f(q_k, \xi_k) \right),$$
$$q_{k+1} = q_k + \alpha_k p_{k+1},$$

where $\text{clip}_r$ is a projection operator that projects the argument onto a ball of radius $r$ at the origin. The algorithm (5) is also reminiscent of Stochastic Primal Averaging (SPA) (Defazio, 2021):

$$p_{k+1} = p_k - \eta_k \nabla f(q_k, \xi_k),$$
$$q_{k+1} = (1 - c_{k+1}) q_k + c_{k+1} p_{k+1}.$$

In Theorem 1 in Defazio (2021) it is shown that this is equivalent to SGD with momentum version

$$p_{k+1} = \beta_k p_k + \nabla f(q_k, \xi_k),$$
$$q_{k+1} = q_k - \alpha_k p_{k+1},$$

if one takes $\eta_{k+1} = \frac{\eta_k - \alpha_k}{\beta_{k+1}}$ and $c_{k+1} = \frac{\alpha_k}{\eta_k}$. The SPA algorithm can be seen as a randomized implicit-explicit Euler discretization of the equation system

$$\dot{p} = -\nabla F(q),$$
$$\dot{q} = p - q,$$

which after a change of variable is equivalent with (6) for $\gamma = 1$. Under the rather strong assumptions that the noise is almost surely bounded (which does not hold for, e.g., Gaussian noise), so-called mixed-clipped SGD with momentum was studied in Zhang et al. (2020a):

$$p_{k+1} = \beta p_k - (1 - \beta) \nabla f(q_k, \xi_k),$$

$$q_{k+1} = q_k - \left[ \nu \min \left( \eta, \frac{\gamma}{\|p_{k+1}\|_2} \right) p_{k+1} + (1 - \nu) \min \left( \eta, \frac{\gamma}{\|\nabla f(q_k, \xi_k)\|_2} \right) \nabla f(q_k, \xi_k) \right],$$

Here, $0 \leq \nu \leq 1$ is an interpolation parameter.

A drawback with the previously mentioned analyses is that the convergence results are obtained in expectation, which means that there is no guarantee that a single path will converge.

## 4.2 HAMILTONIAN DYNAMICS

Hamiltonian dynamics, in its energy conserving form, has been well-explored in the Markov chain Monte Carlo field, compare Leimkuhler & Matthews (2015). In Livingstone et al. (2017), various kinetic energy functions $\varphi$ are considered for equation (8) without the dissipation term $\nabla_{\dot{q}}\mathcal{R}(\dot{q})$.

The algorithm (9) was studied in the context of stochastic differential equations and Langevin dynamics in Stoltz & Trstanova (2018), where the noise is assumed to be Gaussian. In general, this is however a restrictive assumption in the stochastic optimization setting.

The specific update (10) bears resemblance to deterministic time integration- and optimization schemes studied in Franca et al. (2020), that arise as discretizations of the system

$$
\begin{aligned}
\dot{p} &= -\nabla_q H(p,q) - \gamma p, \\
\dot{q} &= \nabla_p H(p,q),
\end{aligned}
\tag{11}
$$

where the dissipation term $\gamma p$ emanates from *Bateman's Lagrangian* $L(q,\dot{q}) = e^{\gamma t}(\varphi^*(\dot{q}) - F(q))$, see Bateman (1931). A similar point of view is also taken in Franca et al. (2021), but where so-called Bregman dynamics is employed. In the (deterministic) optimization setting this was studied in Maddison et al. (2018), where strictly convex kinetic energy functions $\varphi$ are considered. A stochastic gradient version is analysed in Kapoor & Harshvardhan (2021) for strongly convex objective functions $F$.

However, the stochastic optimization algorithm has not been studied for non-convex problems, and an analysis for merely convex (and not strictly convex) kinetic energy functions is lacking.

## 4.3 ALMOST SURE CONVERGENCE

The analysis in this paper is based on the *ODE method*, emanating from Ljung (1976). The particular proof strategy is due to Kushner & Clark (1978), and is based on linear interpolation of the sequence of iterates. The technique was extended to piecewise constant interpolations in Kushner & Yin (2003). The approach relies on the assumption that the iterates generated by the algorithm are finite almost surely; an assumption that has to be verified independently.

A similar analysis of the SGD with momentum was performed in Gadat et al. (2018). It was extended in Barakat et al. (2021), to a class of schemes that encompasses (5). The analytical approach is slightly different and does not cover the normalization- and clipping algorithms that we analyze in this article.

We also note that one can employ an analysis similar to that in e.g. Bottou et al. (2018), along with martingale results like that in Robbins & Siegmund (1971) to obtain almost sure convergence of a subsequence of the iterates. This is for instance the case in Sebbouh et al. (2021) where almost sure convergence guarantees of the type $\min_{0 \le k \le K} \|\nabla F(q_k)\|_2 \to 0$ almost surely for SGD and SGD with momentum are established. These types of results are weaker than those obtained in this paper, since they cannot guarantee that the whole sequence of iterates $\{q_k\}_{k \ge 0}$ converges to a stationary point.

## 4.4 $(L_0, L_1)-$SMOOTHNESS

The $(L_0, L_1)-$smoothness assumption was introduced in Zhang et al. (2020b) as a more appropriate measure of smoothness for certain machine learning problems. It is shown in Zhang et al. (2020b) that the iteration complexity of clipped SGD is bounded, under the assumption that the stochastic gradients are bounded almost surely. The latter is a very restrictive assumption that is not fulfilled even by Gaussian noise. In Zhang et al. (2020a) a clipped algorithm with momentum is shown to converge in expectation to a stationary point under the same strong assumptions on the noise. Similar assumptions are also encountered in e.g. Crawshaw et al. (2022); Li et al. (2024). [1] Koloskova et al. (2023) analyses clipped SGD under Assumption 4.ii), but do not obtain a convergence guarantee due to an *unavoidable bias* (Koloskova et al., 2023). Recently are Wang et al. (2023a) and Faw et al. (2023) obtained convergence guarantees for versions of AdaGrad Norm under the weaker affine

---

[1]Li et al. (2024) also considers the slightly more general case of *sub-Gaussian* noise.

variance-assumption. These results are however only with a certain probability, and there is always some set of positive measure on which the algorithm may not converge.

The convergence guarantees that we obtain in Theorem 5.6 under Assumption 4.i) and Assumption 3.ii) is stronger in the sense that it converges for every path. We also stress the fact that Assumption 3.ii) is relatively weak since it covers all heavy-tailed distributions with finite variance (Rolski et al., 2009). This includes for instance the large class of sub-Weibull distributions, which generalizes sub-Gaussian and sub-exponential distributions (Vladimirova et al., 2020).

## 5 ANALYSIS

We first give a brief overview of the analysis in Section 5.1. In Section 5.2 we describe the setting and in Section 5.3 we give a more detailed outline of the theorems and the proofs. The proofs of the results are given in Appendix A.

### 5.1 BRIEF OVERVIEW

The analysis is split into two parts.

In the first, we show that the iterates of the scheme defined by (9) are finite almost surely, if the objective function $F$ and the convex kinetic energy function $\varphi$ are $L$-smooth and coercive, or if $F$ is $(L_0, L_1)-$smooth and the variance is finite. This is done by constructing a Lyapunov function with the help of the Hamiltonian $H$, and then appealing to the classical Robbins–Siegmund theorem (Robbins & Siegmund, 1971).

In the second, we show that given that the iterates defined by (9) are bounded, they converge almost surely to a stationary point of $F$. We make use of a modification of the *ODE method*, compare Kushner & Yin (2003). Since the scheme is implicit-explicit, we cannot directly apply e.g. Theorem 2.1 in Kushner & Yin (2003). The assumptions that we make on the noise are also much less restrictive that in Kushner & Yin (2003), which in our case would correspond to the stochastic gradients being uniformly bounded in expectation.

Essentially, the idea is to

    i) Introduce a pseudo time $t_k = \sum_{i=0}^{k-1} \alpha_i$ and construct piecewise constant interpolations $P_0(t)$ and $Q_0(t)$ of $\{p_k\}_{k\geq 0}$ and $\{q_k\}_{k\geq 0}$ from (9).

    ii) Show that the time shifted processes $P_k(t) = P_0(t_k + t)$ and $Q_k(t) = Q_0(t_k + t)$ are *equicontinuous in the extended sense* (Kushner & Yin, 2003) and that $P_k(t)$ and $Q_k(t)$ asymptotically satisfies (8).

    iii) At last, make use of the underlying dynamics of (8) to conclude that $\{q_k\}_{k\geq 0}$ converges almost surely to a stationary point of $F$.

### 5.2 SETTING

Let $(\Omega, \mathcal{F}, \mathbb{P})$ be a probability space, and $\{\xi_k\}_{k\geq 0}$ be a sequence of independent, identically distributed random variables. We further let $\mathcal{F}_k$ denote the $\sigma-$algebra generated by $\xi_0, \ldots, \xi_{k-1}$. By $\mathbb{E}_{\xi_k}[X]$ we denote the conditional expectation of a random variable $X$ with respect to $\mathcal{F}_k$. For a set $A \subset \mathbb{R}^d$, we let $N_\delta(A) = \{x : \inf_{a\in A} \|x - a\| < \delta\}$.

#### 5.2.1 BASIC ASSUMPTIONS

We make the following basic assumptions on $f$, $F$ and $\varphi$:

**Assumption 1.** The objective function $F$ is differentiable and satisfies:

    i) *(Coercivity)* $\lim_{\|x\|_2 \to \infty} F(x) = \infty$.

    ii) *(Proper)* There is a number $F_* > -\infty$ such that $F(x) \geq F_*$, $\forall x \in \mathbb{R}^d$.

    iii) *(Locally finite cardinality)* Let $\Lambda = \{q : \nabla F(q) = 0\}$. For every compact set $K \subset \mathbb{R}$, the set $F(\Lambda) \cap K$ has finite cardinality.

Further, the stochastic gradient $\nabla f$ is an unbiased estimator of $\nabla F$, i.e.

    iv) $\mathbb{E}\left[\nabla f(x, \xi)\right] = \nabla F(x)$.

*Remark* 5.1. Assumption 1.i) implies that the sublevel sets $\{x : F(x) \leq c\}$ are bounded, compare Proposition 11.12 in Bauschke & Combettes (2011).

*Remark* 5.2. Assumption 1.iii) is slightly more general than the assumption that $F(\Lambda)$ has finite cardinality, which one often sees; compare e.g. Benaïm (1996). We make use of it in Lemma 5.17, in order to show that the sublevel sets of the Hamiltonian are locally asymptotically stable. Since it is meant to rule out pathological behaviour, it is not obvious how to verify it in practice. However, we note that in many cases the function has isolated equilibria which means that the assumption is satisfied.

**Assumption 2.** The kinetic energy function $\varphi$ is differentiable and satisfies:

    i) *(Lipschitz continuous $\nabla \varphi$)* There is a constant $\lambda > 0$ such that $\|\nabla\varphi(y) - \nabla\varphi(x)\|_2 \leq \lambda\|x - y\|_2$, for all $x, y \in \mathbb{R}^d$.

    ii) *(Convexity)* For all $x, y \in \mathbb{R}^d$, it holds that $\varphi(y) - \varphi(x) \leq \langle \nabla\varphi(y), y - x \rangle$.

    iii) *(Coercivity)* $\lim_{\|x\|_2 \to \infty} \varphi(x) = \infty$.

    iv) *(Proper)* For all $x \in \mathbb{R}^d$, it holds that $\varphi(x) \geq \varphi_* > -\infty$.

*Remark* 5.3. Asumption 1.i), 1.ii), 2.iii) and 2.iv) together implies that the Hamiltonian $H(p, q) = F(q) + \varphi(p)$ is coercive as a function of $q$ and $p$.

In addition to these basic assumptions, we consider three different settings.

### 5.2.2 SETTING 1

In the first setting, $\nabla F$ is Lipschitz continuous but the stochastic gradients can have large variance.

**Assumption 3.** The objective function $F$ and the stochastic gradient $\nabla f$ further satisfy:

    i) *(Lipschitz-continuous $\nabla F$)* There is a constant $L > 0$ such that $\|\nabla F(y) - \nabla F(x)\|_2 \leq L\|x - y\|_2$, for all $x, y \in \mathbb{R}^d$.

    ii) *(Locally bounded variance)* $\mathbb{V}\left[\nabla f(x, \xi)\right] \leq \kappa\left(F(x) - F_*\right) + \tau\|\nabla F(x)\|_2^2 + \sigma^2$,

where $\kappa, \sigma, \tau \geq 0$.

Assumption 3.i) implies that the inequality $F(y) - F(x) \leq \langle \nabla F(x), y - x \rangle + \frac{L}{2}\|x - y\|_2^2$ holds for all $x, y \in \mathbb{R}^d$, compare Lemma 1.2.3 in Nesterov (2018). Assumption 3.ii) was first introduced in Khaled & Richtárik (2020) where it was called as "expected smoothness". It is weak in the sense that it allows for infinite variance in the case that either the gradient or the objective function becomes infinitely large. The condition is similar to e.g. the "affine noise variance" in Wang et al. (2023b) and the "affine variance" in Faw et al. (2023).

### 5.2.3 SETTING 2

Alternatively, we consider the following setting, where we require less regularity of $F$ but instead restrict the variance of the stochastic gradient.

**Assumption 4.** The objective function $F$ and the stochastic gradient $\nabla f$ further satisfy:

    i) *($(L_0, L_1)-$smoothness)* There exists $L_0, L_1$ such that for all $x, y \in \mathbb{R}^d$, if $\|x - y\|_2 \leq \frac{1}{L_1}$, then
$$\|\nabla F(x) - \nabla F(y)\| \leq \left(L_0 + L_1\|\nabla F(y)\|_2\right)\|x - y\|_2.$$

    ii) *(Bounded variance)* $\mathbb{V}\left[\nabla f(x, \xi)\right] \leq \sigma^2$,

    iii) *(Bounded $\nabla \varphi$)* There exists $\Delta > 0$ such that $\|\nabla\varphi(x)\|_2 \leq \Delta$ for all $x \in \mathbb{R}$.

Here $\mathbb{V}\left[\nabla f(x, \xi)\right] = \mathbb{E}\left[\|\nabla f(x, \xi)\|_2^2\right] - \|\nabla F(x)\|_2^2$.

*Remark* 5.4. By *Lyapunov's inequality*, compare p. 230 in Shiryaev (2016), it follows from Assumption 4.ii) that $\mathbb{E}\left[\|\nabla f(x, \xi) - \nabla F(x)\|_2\right] \leq \sigma$.

Assumption 4.ii) sometimes called *heavy-tailed noise* assumption in the literature (Gorbunov et al., 2020; Koloskova et al., 2023). It covers all zero-mean, heavy-tailed distributions with finite second moment, compare Rolski et al. (2009). In particular, it also includes the large class of *sub-Weibull distributions* (Vladimirova et al., 2020), which generalizes random variables of sub-Gaussian and sub-Exponential distribution.

### 5.2.4 SETTING 3

Assumption 4.ii) may be restrictive in some cases, compare (Gurbuzbalaban et al., 2021). Therefore we also consider the the empirical risk minimization setting when the objective function is on the form (2), $f(\cdot, \xi)$ is $(L_0, L_1)-$smooth and $\nabla f(\cdot, \xi)$ is given by (4). In this setting, we can further lower the assumptions on the noise to merely finite expectation:

**Assumption 5.** The objective function satisfies 4.i) and $\nabla \varphi$ satisfies 4.iii). The objective function $F$ and the stochastic gradient $\nabla f$ further satisfy:

  i) *(Empirical risk minimization)* The objective function and the stochastic gradient is of the form (2) and (4), where each for $i$, it holds that $\inf_{q \in \mathbb{R}^d} f_i(q) > -\infty$.

  ii) *(Bounded expectation)* The stochastic gradients satisfy $\mathbb{E}\left[\|\nabla f(x, \xi) - \nabla F(x)\|_2\right] \leq \sigma$.

  iii) *($f(\cdot, \xi)$ is $(L_0, L_1)-$smooth)* The stochastic functions $f(\cdot, \xi)$ are $(L_0, L_1)-$smooth.

*Remark* 5.5. We observe that SGD with momentum, corresponding to (5), requires $\varphi(x) = \|x\|^2/2$, which does not satisfy Assumption 4.iii). It is thus only covered by the analysis in Setting 1. In Setting 2 and 3, we have a weaker regularity assumption on $F$, and this requires us to instead pose stricter requirements on the methods. Overall this indicates that for $(L_0, L_1)-$smooth cost functionals, clipping methods are a better option than SGD.

### 5.2.5 BOOK-KEEPING ASSUMPTIONS

The following assumption on the step sizes $\alpha_k$ is standard and originates from Robbins & Monro (1951). Informally, the step sizes must go to zero in order to counter the stochasticity, but do so slowly enough that we have time to reach a stationary point.

**Assumption 6** (Step sizes). The step size sequence $\{\alpha_k\}_{k \geq 0}$ satisfies $\alpha_0 = 0$ and $\{\alpha_k\}_{k \geq 0} \in \ell^2(\mathbb{R})\backslash\ell^1(\mathbb{R})$.

Our analysis shows convergence to the set of stationary points of $\nabla F$. Under the following additional assumption, we get convergence to a unique stationary point:

**Assumption 7.** The stationary points of $F$ are isolated.

## 5.3 OUTLINE OF PROOF

The proofs of the results in this section can be found in Appendix A. The main theorem is an extension of the approach in Kushner & Yin (2003):

**Theorem 5.6.** *Let Assumptions 1, 2 and 6 be satisfied, as well as either Assumption 3, 4 or 5. Then $\{q_k\}_{k \geq 0}$ converges almost surely to the set of stationary points of the objective function $F$. If we additionally assume that Assumption 7 holds, the convergence is to a unique stationary point.*

The following result is a direct consequence of Theorem 5.6:

**Corollary 5.7** (Convergence in expectation). *Let Assumptions 1, 2, 6 and 7 be valid. Further, let the Hamiltonian be on the form (7) and let the sequences $\{p_k\}_{k \geq 0}$ and $\{q_k\}_{k \geq 0}$ be generated by (9). Then it holds under that*

$$\lim_{k \to \infty} \mathbb{E}\left[\|\nabla F(q_k)\|_2^\theta\right] = 0,$$

*where $\theta = 1$ under Assumption 3 and $\theta = \frac{1}{2}$ under Assumption 4 or 5.*

Our proof strategy consists of two parts. In the first part we show that the sequences $\{p_k\}_{k \geq 0}$ and $\{q_k\}_{k \geq 0}$ are finite almost surely:

**Theorem 5.8** (Finiteness of $\{p_k\}_{k \geq 0}$ and $\{q_k\}_{k \geq 0}$)**.** *Let Assumptions 1, 2 and 6 be valid, as well as either Assumption 3, 4 or 5. Further, let the Hamiltonian be on the form (7) and let the sequences $\{p_k\}_{k \geq 0}$ and $\{q_k\}_{k \geq 0}$ be generated by (9). Then $\{p_k\}_{k \geq 0}$ and $\{q_k\}_{k \geq 0}$ are finite almost surely. Moreover, it holds that $\sup_{k \geq 0} \mathbb{E}\left[F(q_k)\right] < \infty$.*

In the second part of the analysis, we closely follow the ODE method approach as outlined in Kushner & Yin (2003): We start with introducing a pseudo time $t_k = \sum_{i=0}^{k-1} \alpha_i$, and define two piecewise constant, (stochastic) interpolation processes defined by

$$
\begin{aligned}
P_0(t) &= p_0 I_{(-\infty, t_0]}(t) + \sum_{k=0}^{\infty} p_k I_{[t_k, t_{k+1})}(t), \\
Q_0(t) &= q_0 I_{(-\infty, t_0]}(t) + \sum_{k=0}^{\infty} p_k I_{[t_k, t_{k+1})}(t).
\end{aligned}
\tag{12}
$$

We next consider the shifted sequence of processes $\{P_k\}_{k \geq 0}$ and $\{Q_k\}_{k \geq 0}$, defined by

$$
\begin{aligned}
P_k(t) &= P_0(t_k + t), \\
Q_k(t) &= Q_0(t_k + t).
\end{aligned}
\tag{13}
$$

We note that $\{P_k\}_{k \geq 0}$ and $\{Q_k\}_{k \geq 0}$ are stochastic processes; they depend on $\omega \in \Omega$[2] through the stochasticity of the sequences $\{p_k\}_{k \geq 0}$ and $\{q_k\}_{k \geq 0}$. For brevity we will refrain from writing out the dependence on $\omega$.

The next step is to introduce the concept of *extended equicontinuity* (Kushner & Yin, 2003; Freise, 2016):

**Definition 5.9** (Extended equicontinuity)**.** A sequence of $\mathbb{R}^d$-valued functions $\{f_k\}_{k \geq 0}$, defined on $(-\infty, \infty)$, is said to be *equicontinuous in the extended sense* if $\{|f_k(0)|\}_{k \geq 0}$ is bounded and for every $T$ and $\epsilon > 0$ there is $\delta > 0$ such that

$$
\limsup_{k \to \infty} \sup_{0 < |t-s| \leq \delta, \ t,s \in [0,T]} |f_k(t) - f_k(s)| \leq \epsilon.
\tag{14}
$$

Following Freise (2016), we show that the process $\{Z_k\}_{k \geq 0} = \{(P_k, Q_k)\}_{k \geq 0}$, where $\{P_k\}_{k \geq 0}$ and $\{Q_k\}_{k \geq 0}$ defined by (13), is equicontinuous in the extended sense:

**Lemma 5.10** (Equicontinuous in the extended sense)**.** *Consider $\{Z_k\}_{k \geq 0} = \{(P_k, Q_k)\}_{k \geq 0}$ where the sequences $\{P_k\}_{k \geq 0}$ and $\{Q_k\}_{k \geq 0}$ are defined by (13) (equivalently, by (37)). Suppose that $\{p_k\}_{k \geq 0}$ and $\{q_k\}_{k \geq 0}$ are defined by (9), and that the Hamiltonian is on the form (7). Further, let Assumptions 1, 2 and 6 be valid, as well as either Assumption 3, 4 or 5. Then $\{Z_k\}_{k \geq 0}$ is equicontinuous in the extended sense, almost surely.*

We can then appeal to the *extended/discontinuous Arzelà–Ascoli theorem* (Kushner & Yin, 2003; Freise, 2016; Droniou & Eymard, 2016), to conclude that $\{Z_k\}_{k \geq 0}$ has a subsequence that converges to a continuous function $z$:

**Theorem 5.11** (Discontinuous Arzelà–Ascoli theorem)**.** *Let $\{f_k\}_{k \geq 0}$ be a sequence of functions, defined on $\mathbb{R}^d$, that is equicontinuous in the extended sense. Then there is a subsequence $\{f_{n_k}\}_{n_k \geq 0}$ of $\{f_k\}_{k \geq 0}$, that converges uniformly on compact sets to a continuous function.*

For a proof see, e.g. Theorem 6.2 in Droniou & Eymard (2016) or Theorem 12.3 in Billingsley (1968).

With this established, we proceed to show that $\{Z_k\}_{k \geq 0}$ is an *asymptotic solution*[3] to (8); i.e. asymptotically $\{P_k\}_{k \geq 0}$ and $\{Q_k\}_{k \geq 0}$ satisfy (8). More precisely we show

---

[2]Here $\omega$ is an *outcome* and $\Omega$ is the *sample space* of the underlying probability space $(\Omega, \mathcal{F}, \mathbb{P})$.

[3]By *Grönwall's inequality* (compare e.g. Ethier & Kurtz (1986)) this is equivalent to (12) being an *asymptotic pseudotrajectory* (Benaïm, 1999) to (8).

**Lemma 5.12** (Asymptotic solutions). *With the same assumptions and notation as in Lemma 5.10, we can write*

$$P_k(t) = P_k(0) - \int_0^t \nabla F(Q_k(s))\mathrm{d}s - \gamma \int_0^t \nabla \varphi(P_k(s))\mathrm{d}s + M_k(t) + \mu_k(t),$$

$$Q_k(t) = Q_k(0) + \int_0^t \nabla \varphi(P_k(s))\mathrm{d}s + \nu_k(t) + \kappa_k(t), \tag{15}$$

*where the functions $\{M_k\}_{k\geq 0}$, $\{\mu_k\}_{k\geq 0}$, $\{\nu_k\}_{k\geq 0}$ and $\{\kappa_k\}_{k\geq 0}$ converge to $0$ uniformly on compact sets almost surely.*

It follows that any limit point of $\{Z_k\}_{k\geq 0}$ satisfies

$$P(t) = P(0) - \int_0^t \nabla F(Q(s))\mathrm{d}s - \gamma \int_0^t \nabla \varphi(P(s))\mathrm{d}s$$

$$Q(t) = Q(0) + \int_0^t \nabla \varphi(P(s))\mathrm{d}s. \tag{16}$$

The limits we can extract by appealing to Theorem 5.11 are continuous. Thus it follows from (16) and the fundamental theorem of calculus that they are differentiable and satisfy (8).

*Remark* 5.13. The functions $\{\mu_k\}_{k\geq 0}$ and $\{\nu_k\}_{k\geq 0}$ are essentially what is left when we have rewritten the sums in (12) as integrals. The functions $\{M_k\}_{k\geq 0}$ account for the difference between $\nabla F(q_k)$ and $\nabla f(q_k, \xi_k)$ and $\kappa_k(t)$ for the implicit discretization in the second equation of (9).

*Remark* 5.14. The convergence "uniformly on compact sets almost surely" is to be understood as uniformly on compact sets in $t$ and almost surely in $\omega$. For example, for the sequence $\{M_k\}_{k\geq 0}$ we have that for any compact set $K \subset \mathbb{R}$, $\lim_{k\to\infty} \sup_{t\in K} \|M_k(t)(\omega)\|_2 = 0$ for almost all $\omega \in \Omega$.

We recall the definition of a *locally asymptotically stable set* (Borkar, 2008; Kushner & Yin, 2003):

**Definition 5.15** (Locally asymptotically stable set). A set $A$ is said to be *Lyapunov stable* if for any $\epsilon > 0$, there exists a $\delta > 0$ such that every trajectory initiated in the $N_\delta(A)$ remains in $N_\epsilon(A)$. It is *locally asymptotically stable* if every such path ultimately goes to $A$.

With this in mind, we show the following theorem, which is essentially an adaptation of Theorem 5.2.1 in Kushner & Yin (2003).

**Theorem 5.16.** *Under the same assumptions and notation as in Theorem 5.6, let $A$ be a locally asymptotically stable set for (8). If there exists a compact set in the domain of attraction of $A$ that $\{z_k\}_{k\geq 0}$ visits infinitely often, then $z_k \to A$ almost surely:*

$$\lim_{k\to\infty} \inf_{a\in A} \|z_k - a\|_{\ell^2(\mathbb{R}^{2d})} = 0, \quad a.s. \tag{17}$$

The next step is to prove the following, which gives us specific locally asymptotically stable sets:

**Lemma 5.17.** *Consider the same assumptions and notation as in Theorem 5.6. For each $c$, if the set $\{z : H(z) \leq c\}$ is non-empty, it is a locally asymptotically stable set for the solutions to (8).*

In particular, the set $A = \{z : H(z) \leq \liminf_k H(z_k)\}$ is locally asymptotically stable. By the properties of $\liminf$, we can also find a compact set which $z_k$ enters infinitely often, and we can therefore apply Theorem 5.16 to conclude that $z_k \to A$. The final step is to show that this convergence in fact implies convergence to the set of stationary points of $H$, and therefore that $q_k$ converges to the set of stationary points of $F$. Under Assumption 7 we can additionally conclude that the convergence is to a unique equilibrium.

# 6 CONCLUSIONS

In this paper, we have shown that the stochastic Hamiltonian descent algorithm (9), arising as a stochastic explicit-implicit Euler discretization of (8), under weak assumptions converges almost surely to the set of stationary points of the objective function $F$. In the terminology of Robbins & Monro (1951), this means that the estimator determined by $\{q_k\}$ is a strongly consistent estimator of a stationary point of $F$. (Here, the designated asymptoticity is with respect to the number of iterations instead of the sample size.) Similarly, the result in Corollary 5.7 is akin to $\{q_k\}_{k\geq 0}$ being an asymptotically unbiased estimator of a stationary point $q_*$.

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

## A    ANALYSIS

As explained in Section 5.3, the two main steps of the convergence analysis are to first prove that $p_k$ and $q_k$ are finite almost surely, and then to use this a priori result to show that they in fact converge.

## A.1 THE SEQUENCES $\{p_k\}_{k\geq 0}$ AND $\{q_k\}_{k\geq 0}$ ARE FINITE ALMOST SURELY

We first prove Theorem 5.8:

**Theorem 5.8** (Finiteness of $\{p_k\}_{k\geq 0}$ and $\{q_k\}_{k\geq 0}$). *Let Assumptions 1, 2 and 6 be valid, as well as either Assumption 3, 4 or 5. Further, let the Hamiltonian be on the form (7) and let the sequences $\{p_k\}_{k\geq 0}$ and $\{q_k\}_{k\geq 0}$ be generated by (9). Then $\{p_k\}_{k\geq 0}$ and $\{q_k\}_{k\geq 0}$ are finite almost surely. Moreover, it holds that $\sup_{k\geq 0} \mathbb{E}\left[F(q_k)\right] < \infty$.*

The proof relies on the Robbins-Siegmund theorem:

**Theorem A.1** ((Robbins & Siegmund, 1971)). *Let $(\Omega, \mathcal{F}, \mathbb{P})$ be a probability space and $\mathcal{F}_1 \subset \mathcal{F}_2 \subset \dots$ be a sequence of sub-$\sigma$-algebras of $\mathcal{F}$. For each $k = 1, 2, \dots$ let $V_k, \beta_k, X_k$ and $Y_k$ be non-negative $\mathcal{F}_k$-measurable random variables such that*

$$\mathbb{E}\left[V_{k+1}|\mathcal{F}_k\right] \leq V_k(1 + \beta_k) + X_k - Y_k.$$

*Then*

$$\lim_{k\to\infty} V_k = V \tag{18}$$

*exists and is finite and $\sum_k Y_k < \infty$ on the set*

$$\left\{\omega : \sum_k \beta_k < \infty, \sum_k X_k < \infty\right\}.$$

We first consider Setting 1, i.e. with Assumption 3. We note that $V = H(p, q) - F_* - \varphi_*$ is a Lyapunov function, since the system is nearly Hamiltonian; $\dot{V}(t) = \langle\nabla_p H(p, q), \dot{p}\rangle + \langle\nabla_q H(p, q), \dot{q}\rangle = -\gamma\langle\nabla_p H(p, q), \nabla_p H(p, q)\rangle < 0$. The strategy is now to introduce a corresponding (almost) discrete Lyapunov function $V_k = H(p_k, q_k) - F_* - \varphi_* = F(q_k) - F_* + \varphi(p_k) - \varphi_*$. We can then use $L$-smoothness of $F$ and the convexity of $\varphi$ to bound the difference $V_{k+1} - V_k$ by $\alpha_k^2 V_k$ plus higher-order terms of $\alpha_k$, whereupon we can appeal to Theorem A.1 to conclude that $\{p_k\}_{k\geq 1}$ and $\{q_k\}_{k\geq 1}$ are finite a.s.

*Proof of Theorem 5.8 in Setting 1.* Let $V_k = H(p_k, q_k) - F_* - \varphi_*$. Then we have that

$$V_{k+1} - V_k = F(q_{k+1}) - F(q_k) + \varphi(p_{k+1}) - \varphi(p_k). \tag{19}$$

By $L$-smoothness of $F$ and convexity of $\varphi$, this is less than or equal to

$$\langle\nabla F(q_k), q_{k+1} - q_k\rangle + \frac{L}{2}\|q_{k+1} - q_k\|_2^2 + \langle\nabla\varphi(p_{k+1}), p_{k+1} - p_k\rangle.$$

We insert (9) into the previous expression to obtain that it is equal to

$$\alpha_k\langle\nabla F(q_k) - \nabla f(q_k, \xi_k), \nabla\varphi(p_{k+1})\rangle + \frac{L\alpha_k^2}{2}\|\nabla\varphi(p_{k+1})\|_2^2 - \alpha_k\gamma\langle\nabla\varphi(p_{k+1}), \nabla\varphi(p_k)\rangle$$

$$=: I_1 + I_2 + I_3.$$

We add and subtract $\alpha_k\langle\nabla F(q_k) - \nabla f(q_k, \xi_k), \nabla\varphi(p_k)\rangle$ to the first term:

$$I_1 = \alpha_k\langle\nabla F(q_k) - \nabla f(q_k, \xi_k), \nabla\varphi(p_k)\rangle + \alpha_k\langle\nabla F(q_k) - \nabla f(q_k, \xi_k), \nabla\varphi(p_{k+1}) - \nabla\varphi(p_k)\rangle.$$

When we take the conditional expectation (w.r.t. the sigma algebra generated by $\xi_1, \dots, \xi_{k-1}$) of $I_1$, the first term is 0 by the unbiasedness of the gradient and the independence of $\{\xi_k\}$:

$$\mathbb{E}_{\xi_k}\left[I_1\right] = \alpha_k\mathbb{E}_{\xi_k}\left[\langle\nabla F(q_k) - \nabla f(q_k, \xi_k), \nabla\varphi(p_{k+1}) - \nabla\varphi(p_k)\rangle\right].$$

Using Assumption 2.i), we can bound $I_3$ as

$$I_3 = -\alpha_k\gamma\langle\nabla\varphi(p_{k+1}), \nabla\varphi(p_k)\rangle \leq \frac{\alpha_k\gamma}{2}\|\nabla\varphi(p_{k+1}) - \nabla\varphi(p_k)\|_2^2 \leq \frac{\alpha_k\gamma\lambda^2}{2}\|p_{k+1} - p_k\|_2^2.$$

After taking the expectation of (19), we thus get the bound

$$\mathbb{E}_{\xi_k}\left[V_{k+1}\right] - V_k \leq \alpha_k\mathbb{E}_{\xi_k}\left[\langle\nabla F(q_k) - \nabla f(q_k, \xi_k), \nabla\varphi(p_{k+1}) - \nabla\varphi(p_k)\rangle\right]$$

$$+ \frac{L\alpha_k^2}{2}\mathbb{E}_{\xi_k}\left[\|\nabla\varphi(p_{k+1})\|_2^2\right] + \frac{\alpha_k\gamma\lambda^2}{2}\mathbb{E}_{\xi_k}\left[\|p_{k+1} - p_k\|_2^2\right] \tag{20}$$

$$=: I_1' + I_2 + I_3'.$$

We now make use of Cauchy–Schwarz inequality along with the Lipschitz continuity of $\nabla\varphi$ to bound $I_1'$ as

$$I_1' \leq \alpha_k \lambda \mathbb{E}_{\xi_k} \left[ \|\nabla F(q_k) - \nabla f(q_k, \xi_k)\|_2 \|p_{k+1} - p_k\|_2 \right].$$

We insert (9) into the previous expression, and make use of Young's inequality for products, $ab \leq \frac{a^2}{2} + \frac{b^2}{2}$, to obtain that

$$I_1' \leq \alpha_k^2 \lambda \mathbb{E}_{\xi_k} \left[ \|\nabla F(q_k) - \nabla f(q_k, \xi_k)\|_2 \|\nabla f(q_k, \xi_k) + \gamma \nabla\varphi(p_k)\|_2 \right]$$
$$\leq \frac{\alpha_k^2}{2} \lambda \mathbb{E}_{\xi_k} \left[ \|\nabla F(q_k) - \nabla f(q_k, \xi_k)\|_2^2 \right] + \frac{\alpha_k^2}{2} \lambda \mathbb{E}_{\xi_k} \left[ \|\nabla f(q_k, \xi_k) + \gamma \nabla\varphi(p_k)\|_2^2 \right].$$

Making use of the inequality

$$\|x - y\|_2^2 \leq 2\|x\|_2^2 + 2\|y\|_2^2, \tag{21}$$

we can further bound $I_1'$ by

$$I_1' \leq \frac{\alpha_k^2}{2} \lambda \mathbb{E}_{\xi_k} \left[ \|\nabla F(q_k) - \nabla f(q_k, \xi_k)\|_2^2 \right]$$
$$+ \alpha_k^2 \lambda \mathbb{E}_{\xi_k} \left[ \|\nabla f(q_k, \xi_k)\|_2^2 \right] + \alpha_k^2 \lambda \gamma^2 \mathbb{E}_{\xi_k} \left[ \|\nabla\varphi(p_k)\|_2^2 \right].$$

At last we make use of Assumption 3.ii) to get that

$$I_1' \leq \frac{\alpha_k^2}{2} \lambda \left( \kappa(F(q_k) - F_*) + \tau \|\nabla F(q_k)\|_2^2 + \sigma^2 \right)$$
$$+ \alpha_k^2 \lambda \left( \kappa(F(q_k) - F_*) + (1 + \tau)\|\nabla F(q_k)\|_2^2 + \sigma^2 \right) + \alpha_k^2 \lambda \gamma^2 \mathbb{E}_{\xi_k} \left[ \|\nabla\varphi(p_k)\|_2^2 \right].$$

We now turn our attention to the term $I_2$ in (20). Adding and subtracting $\nabla\varphi(p_k)$ and making use of Assumption 2.i) we get that

$$I_2 \leq \frac{L\alpha_k^2}{2} \mathbb{E}_{\xi_k} \left[ \|\nabla\varphi(p_{k+1}) - \nabla\varphi(p_k)\|_2^2 \right] + \frac{L\alpha_k^2}{2} \mathbb{E}_{\xi_k} \left[ \|\nabla\varphi(p_k)\|_2^2 \right]$$
$$\leq \frac{L\lambda^2\alpha_k^2}{2} \mathbb{E}_{\xi_k} \left[ \|p_{k+1} - p_k\|_2^2 \right] + \frac{L\alpha_k^2}{2} \|\nabla\varphi(p_k)\|_2^2$$
$$\leq L\alpha_k^4 \lambda^2 \mathbb{E}_{\xi_k} \left[ \|\nabla f(q_k, \xi_k)\|_2^2 \right] + \left( L\alpha_k^4 \gamma^2 \lambda^2 + \frac{L\alpha_k^2}{2} \right) \|\nabla\varphi(p_k)\|_2^2,$$

where we have made use of (9) and (21) in the last step. Making use of Assumption 2.i) again we obtain that

$$I_2 \leq L\alpha_k^4 \lambda^2 \left( \kappa(F(q_k) - F_*) + (1 + \tau)\|\nabla F(q_k)\|_2^2 + \sigma^2 \right) + \left( L\alpha_k^3 \gamma^2 \lambda^2 + \frac{L\alpha_k^2}{2} \right) \|\nabla\varphi(p_k)\|_2^2.$$

In a similar way, we find that

$$I_3' \leq \alpha_k^3 \gamma \lambda^2 \left( \kappa(F(q_k) - F_*) + (1 + \tau)\|\nabla F(q_k)\|_2^2 + \sigma^2 \right) + \alpha_k^3 \gamma^3 \lambda^2 \|\nabla\varphi(p_k)\|_2^2.$$

Gathering up the terms, we get that

$$\mathbb{E}_{\xi_k} [V_{k+1}] - V_k \leq \left( \frac{\alpha_k^2 \lambda}{2} + \alpha_k^2 \lambda + L\alpha_k^4 \lambda^2 + \alpha_k^3 \gamma \lambda^2 \right) \sigma^2$$
$$+ \alpha_k^2 \kappa \lambda \left( \frac{3}{2} + L\alpha_k^2 \lambda + \alpha_k \gamma \lambda \right) (F(q_k) - F_*)$$
$$+ \left( \frac{\alpha_k^2 \lambda \tau}{2} + \alpha_k^2 \lambda(1 + \tau) + L\alpha_k^4 \lambda^2(1 + \tau) + \alpha_k^3 \gamma \lambda^2(1 + \tau) \right) \|\nabla F(q_k)\|_2^2$$
$$+ \left( \alpha_k^2 \gamma^2 \lambda + L\alpha_k^4 \lambda^2 \gamma^2 + \frac{L\alpha_k^2}{2} + \alpha_k^3 \gamma^3 \lambda^2 \right) \|\nabla\varphi(p_k)\|_2^2.$$

Making use of Lemma B.6, we see that

$$
\begin{aligned}
\mathbb{E}_{\xi_k} & \left[ V_{k+1} \right] - V_k \\
& \leq \left( \frac{\alpha_k^2 \lambda}{2} + \alpha_k^2 \lambda + L\alpha_k^4 \lambda^2 + \alpha_k^3 \gamma \lambda^2 \right) \sigma^2 \\
& \quad + \left( \alpha_k^2 \kappa \lambda \left( \frac{3}{2} + L\alpha_k^2 \lambda + \alpha_k \gamma \lambda \right) \right. \\
& \quad + 2L \left( \frac{\alpha_k^2 \lambda \tau}{2} + \alpha_k^2 \lambda (1+\tau) + L\alpha_k^4 \lambda^2 (1+\tau) + \alpha_k^3 \gamma \lambda^2 (1+\tau) \right) \right) (F(q_k) - F_*) \\
& \quad + \left( \alpha_k^2 \gamma^2 \lambda + L\alpha_k^4 \lambda^2 \gamma^2 + \frac{L\alpha_k^2}{2} + \alpha_k^3 \gamma^3 \lambda^2 \right) (\varphi(p_k) - \varphi_*).
\end{aligned}
\tag{22}
$$

Now define

$$
C_1(\alpha_k) = \sigma^2 \left( \frac{\alpha_k^2 \lambda}{2} + \alpha_k^2 \lambda + L\alpha_k^4 \lambda^2 + \alpha_k^3 \gamma \lambda^2 \right)
$$

and let $C_2(\alpha_k)$ be the maximum of the terms in front of $F(q_k) - F_*$ and $\varphi(p_k) - \varphi_*$. It follows that

$$
\mathbb{E}_{\xi_k} \left[ V_{k+1} \right] - V_k \leq C_1(\alpha_k) + C_2(\alpha_k) V_k.
\tag{23}
$$

Since $C_1$ and $C_2$ only contain second-order terms of $\alpha_k$ (and by assumption $\sum_{k=1}^{\infty} \alpha_k^2 < \infty$), we have that

$$
\sum_{k=0}^{\infty} C_1(\alpha_k) < \infty, \ \sum_{k=0}^{\infty} C_2(\alpha_k) < \infty.
$$

We can thus make use of the Robbins–Siegmund theorem with $\beta_k = C_2(\alpha_k)$, $X_k = C_1(\alpha_k)$ and $Y_k = 0$ to conclude that $V_k$ tends to a non-negative, finite, random variable $V$ almost surely. Since $F$ and $\varphi$ are assumed to be coercive, this implies that $\{p_k\}_{k \geq 1}$ and $\{q_k\}_{k \geq 1}$ are finite almost surely. For the second claim of the proof, we define

$$
S_k = \frac{V_k}{\prod_{j=0}^{k-1} (1 + C_2(\alpha_j))}.
$$

By (23), we have that

$$
\mathbb{E}_{\xi_k} \left[ S_{k+1} \right] \leq S_k + \frac{C_1(\alpha_k)}{\prod_{j=0}^{k} (1 + C_2(\alpha_j))} \leq S_k + C_1(\alpha_k).
$$

Taking the expectation and, summing from 0 to $K - 1$, we see that

$$
\mathbb{E} \left[ S_K \right] \leq S_0 + \sum_{k=0}^{K-1} C_1(\alpha_k).
$$

We multiply both sides of the previous inequality with $\prod_{k=0}^{K-1} (1 + C_2(\alpha_k))$

$$
\begin{aligned}
\mathbb{E} \left[ V_K \right] & \leq \prod_{k=0}^{K-1} (1 + C_2(\alpha_k)) \left( S_0 + \sum_{k=0}^{K-1} C_1(\alpha_k) \right) \\
& \leq e^{\sum_{j=0}^{K-1} C_2(\alpha_k))} \cdot \left( S_0 + \sum_{k=0}^{K-1} C_1(\alpha_k) \right),
\end{aligned}
$$

where we have used the fact that $1 + x \leq e^x$ in the second step. Letting $K$ tend to infinity on the left hand side and using the fact that $\sum_{k=0}^{\infty} C_2(\alpha_k)) < \infty$ we see that the last claim of the theorem also holds:

$$
\sup_k \mathbb{E} \left[ V_k \right] < \infty.
$$

$\square$

We now give a proof of Theorem 5.8 in Setting 2, i.e. with Assumption 4.

*Proof of Theorem 5.8 in Setting 2.* By Assumption 4.i) it holds for $\|x_k - x_{k+1}\| \le \frac{1}{L_1}$ that

$$F(x_{k+1}) - F(x_k) \le \langle \nabla F(x_k), x_{k+1} - x_k \rangle + \frac{L_0 + L_1 \|\nabla F(x_k)\|}{2} \|x_{k+1} - x_k\|^2. \qquad (24)$$

Since

$$q_{k+1} - q_k = \alpha_k \nabla \varphi(p_{k+1}) \qquad (25)$$

we get for large enough $k$ that

$$\|q_{k+1} - q_k\| = \alpha_k \|\nabla \varphi(p_{k+1})\| \le \alpha_k \Delta \le \frac{1}{L_1}, \qquad (26)$$

by Assumption 4.iii). If we insert (25) into (24), we get

$$F(q_{k+1}) - F(q_k) \le \langle \nabla F(q_k), q_{k+1} - q_k \rangle + \frac{L_0}{2} \|q_{k+1} - q_k\|^2 + \frac{L_1}{2} \|\nabla F(q_k)\| \|q_{k+1} - q_k\|^2. \qquad (27)$$

By (26), we get that

$$F(q_{k+1}) - F(q_k) \le \alpha_k \langle \nabla F(q_k), \nabla \varphi(p_{k+1}) \rangle + \frac{L_0}{2} \alpha_k^2 \Delta^2 + \frac{L_1}{2} \|\nabla F(q_k)\| \alpha_k^2 \Delta^2. \qquad (28)$$

By Assumption 2.ii) we have that

$$\varphi(p_{k+1}) - \varphi(p_k) \le \langle \nabla \varphi(p_{k+1}), p_{k+1} - p_k \rangle$$
$$= -\alpha_k \langle \nabla \varphi(p_{k+1}), \nabla f(q_k, \xi_k) \rangle - \alpha_k \gamma \langle \nabla \varphi(p_{k+1}), \nabla \varphi(p_k) \rangle$$

With $H(p, q) = F(q) + \varphi(p)$ and $V_k = H(p_k, q_k) - F_* - \varphi_*$ as in the previous proof we thus get that

$$V_{k+1} - V_k \le \alpha_k \langle \nabla F(q_k), \nabla \varphi(p_{k+1}) \rangle + \frac{L_0}{2} \alpha_k^2 \Delta^2 + \frac{\alpha_k^2 L_1}{2} \|\nabla F(q_k)\| \Delta^2$$
$$- \alpha_k \langle \nabla \varphi(p_{k+1}), \nabla f(q_k, \xi_k) \rangle - \alpha_k \gamma \langle \nabla \varphi(p_{k+1}), \nabla \varphi(p_k) \rangle,$$

which can be rewritten as

$$V_{k+1} - V_k \le \alpha_k \langle \nabla F(q_k) - \nabla f(q_k, \xi_k), \nabla \varphi(p_{k+1}) \rangle + \frac{L_0}{2} \alpha_k^2 \Delta^2 + \frac{\alpha_k^2 L_1}{2} \|\nabla F(q_k)\| \Delta^2$$
$$- \alpha_k \gamma \langle \nabla \varphi(p_{k+1}), \nabla \varphi(p_k) \rangle.$$

We add and subtract $\nabla \varphi(p_k)$ in the first scalar product:

$$V_{k+1} - V_k \le \alpha_k \langle \nabla F(q_k) - \nabla f(q_k, \xi_k), \nabla \varphi(p_{k+1}) - \nabla \varphi(p_k) \rangle$$
$$+ \alpha_k \langle \nabla F(q_k) - \nabla f(q_k, \xi_k), \nabla \varphi(p_k) \rangle \qquad (29)$$
$$+ \frac{L_0}{2} \alpha_k^2 \Delta^2 + \frac{\alpha_k^2 L_1}{2} \|\nabla F(q_k)\| \Delta^2 - \alpha_k \gamma \langle \nabla \varphi(p_{k+1}), \nabla \varphi(p_k) \rangle.$$

The second scalar product disappears due to the unbiasedness of $\nabla f(q_k, \xi_k)$ and the fact that $\xi_k$ is independent of $q_k$ and $p_k$. We now focus on the first scalar product in (29). Taking the conditional expectation and using Cauchy–Schwarz inequality, we see that

$$\alpha_k \mathbb{E}_{\xi_k} \left[ \langle \nabla F(q_k) - \nabla f(q_k, \xi_k), \nabla \varphi(p_{k+1}) - \nabla \varphi(p_k) \rangle \right]$$
$$\le \alpha_k \mathbb{E}_{\xi_k} \left[ \|\nabla F(q_k) - \nabla f(q_k, \xi_k)\| \|\nabla \varphi(p_{k+1}) - \nabla \varphi(p_k)\| \right]$$

Using the Lipschitz continuity of $\nabla \varphi$, this can be further bounded as

$$\alpha_k^2 \lambda \mathbb{E}_{\xi_k} \left[ \|\nabla F(q_k) - \nabla f(q_k, \xi_k)\| \|\nabla f(q_k, \xi_k) - \gamma \nabla \varphi(p_k)\| \right]$$
$$\le \alpha_k^2 \lambda \mathbb{E}_{\xi_k} \left[ \|\nabla F(q_k) - \nabla f(q_k, \xi_k)\| \left( \|\nabla f(q_k, \xi_k)\| + \gamma \|\nabla \varphi(p_k)\| \right) \right]. \qquad (30)$$

We now add and subtract $\nabla F(q_k)$ inside the $\|\nabla f(q_k, \xi_k)\|-$term and make use of the triangle inequality to bound the previous expression by

$$\alpha_k^2 \lambda \mathbb{E}_{\xi_k} \left[ \|\nabla F(q_k) - \nabla f(q_k, \xi_k)\| \left( \|\nabla F(q_k) - \nabla f(q_k, \xi_k)\| + \|\nabla F(q_k)\| + \gamma\Delta \right) \right]$$
$$= \alpha_k^2 \lambda \mathbb{E}_{\xi_k} \left[ \|\nabla F(q_k) - \nabla f(q_k, \xi_k)\|^2 + \|\nabla F(q_k) - \nabla f(q_k, \xi_k)\| \|\nabla F(q_k)\| \right.$$
$$\left. + \gamma\Delta \|\nabla F(q_k) - \nabla f(q_k, \xi_k)\| \right]$$

where we also have used the assumption that $\|\nabla\varphi(p_k)\| \leq \Delta$. Now, the first term can by Assumption 4.ii) be bounded by

$$\mathbb{E}_{\xi_k} \left[ \|\nabla F(q_k) - \nabla f(q_k, \xi_k)\|^2 \right] \leq \sigma^2.$$

By Remark 5.4, we can bound the second term as follows

$$\mathbb{E}_{\xi_k} \left[ \|\nabla F(q_k) - \nabla f(q_k, \xi_k)\| \|\nabla F(q_k)\| \right] \leq \sigma \|\nabla F(q_k)\|,$$

since $\xi_k$ is independent of $\|\nabla F(q_k)\|$. Likewise, we can bound the last expectation by $\sigma$. Thus, we arrive at the bound

$$\alpha_k \mathbb{E}_{\xi_k} \left[ \langle \nabla F(q_k) - \nabla f(q_k, \xi_k), \nabla\varphi(p_{k+1}) - \nabla\varphi(p_k) \rangle \right] \leq \alpha_k^2 \lambda \left( \sigma^2 + \sigma\|\nabla F(q_k)\| + \gamma\Delta\sigma \right). \tag{31}$$

We can bound the last inner product of (29) using Lemma B.1 in Zhang et al. (2020a), taking $\mu = 0$:

$$-\alpha_k \gamma \langle \nabla\varphi(p_{k+1}), \nabla\varphi(p_k) \rangle \leq -\alpha_k \gamma \|\nabla\varphi(p_k)\|^2 + \alpha_k \gamma \|\nabla\varphi(p_{k+1}) - \nabla\varphi(p_k)\| \|\nabla\varphi(p_k)\|$$
$$\leq \alpha_k^2 \gamma \lambda 2 \|\nabla f(q_k, \xi_k) - \gamma\nabla\varphi(p_k)\|\Delta.$$

By Remark 5.4 we thus get

$$-\alpha_k \gamma \mathbb{E}_{\xi_k} \left[ \langle \nabla\varphi(p_{k+1}), \nabla\varphi(p_k) \rangle \right]$$
$$\leq \alpha_k^2 \gamma \lambda \Delta \left( \mathbb{E}_{\xi_k} \left[ \|\nabla f(q_k, \xi_k)\| \right] + \gamma\|\nabla\varphi(p_k)\| \right)$$
$$\leq \alpha_k^2 \gamma \lambda \Delta \left( \mathbb{E}_{\xi_k} \left[ \|\nabla f(q_k, \xi_k) - \nabla F(q_k)\| \right] + \|\nabla F(q_k)\|_2 + \gamma\|\nabla\varphi(p_k)\| \right) \tag{32}$$
$$\leq \alpha_k^2 \gamma \lambda \Delta \left( \sigma + \|\nabla F(q_k)\|_2 + \gamma\Delta \right).$$

Inserting (31) and (32) into (29), we get that

$$\mathbb{E}_{\xi_k} \left[ V_{k+1} \right] - V_k \leq \alpha_k^2 \lambda \left( \sigma^2 + \sigma\|\nabla F(q_k)\| + \gamma\Delta\sigma \right) + \frac{L_0}{2} \alpha_k^2 \Delta^2$$
$$+ \frac{\alpha_k^2 L_1}{2} \|\nabla F(q_k)\| \Delta^2 + \alpha_k^2 \gamma \lambda \Delta \left( \sigma + \|\nabla F(q_k)\|_2 + \gamma\Delta \right) \tag{33}$$

From Lemma B.6, we can bound the $\|\nabla F(q_k)\|-$terms, and obtain the bound

$$\mathbb{E}_{\xi_k} \left[ V_{k+1} \right] - V_k \leq \alpha_k^2 \lambda \left( \sigma^2 + \sigma \left( 2L_1(F(q) - F_*) + \frac{L_0}{L_1} \right) + \gamma\Delta\sigma \right) + \frac{L_0}{2} \alpha_k^2 \Delta^2$$
$$+ \frac{\alpha_k^2 L_1}{2} \left( 2L_1(F(q) - F_*) + \frac{L_0}{L_1} \right) \Delta^2 \tag{34}$$
$$+ \alpha_k^2 \gamma \lambda \Delta \left( \sigma + \left( 2L_1(F(q) - F_*) + \frac{L_0}{L_1} \right) + \gamma\Delta \right).$$

We now define

$$C_1(\alpha_k) = \alpha_k^2 \lambda \sigma 2L_1 + \alpha_k^2 L_1^2 \Delta^2 + \alpha_k^2 \gamma \lambda \Delta 2L_1,$$

$$C_2(\alpha_k) = \alpha_k^2 \lambda \sigma^2 + \alpha_k^2 \lambda \sigma \frac{L_0}{L_1} + \alpha_k^2 \lambda \gamma \Delta \sigma + \alpha_k^2 L_0 \Delta^2 + \alpha_k^2 \gamma \lambda \Delta \sigma + \alpha_k^2 \gamma \lambda \Delta \frac{L_0}{L_1} + \alpha_k^2 \gamma^2 \lambda \Delta^2.$$

We see that

$$\mathbb{E}_{\xi_k} \left[ V_{k+1} \right] - V_k \leq C_1(\alpha_k)(F(q_k) - F_*) + C_2(\alpha_k)$$
$$\leq C_1(\alpha_k)(H(p_k, q_k) - F_* - \varphi_*) + C_2(\alpha_k), \tag{35}$$

where we have used the fact that $\varphi(p_k) - \varphi_* \geq 0$. Since $\sum_{k \geq 0} C_i(\alpha_k) < \infty$ for $i = 1, 2$, we can appeal to the Robbins–Siegmund theorem to conclude that $\lim_{k \to \infty} V_k$ exists and is finite almost surely. Since $F$ and $\varphi$ are coercive this implies that $\sup_k \|p_k\| < \infty$ and $\sup_k \|q_k\| < \infty$ almost surely.

$\square$

*Proof of Theorem 5.8 in Setting 3.* Let $F_*$ be as in Lemma B.7. By Lemma B.7, we have that the right-hand side of (30) can be bounded by

$$\alpha_k^2 \lambda \mathbb{E}_{\xi_k} \left[ \|\nabla F(q_k) - \nabla f(q_k, \xi_k)\| \right] (2L_1 N(F(q_k) - F_*) + \gamma \Delta).$$

By Assumption 5.ii) this can in its turn be bounded by

$$\alpha_k^2 \lambda \sigma \left( 2L_1 N(F(q_k) - F_*) + \gamma \Delta \right).$$

The rest of the proof proceeds exactly like that of Setting 2 (with suitable modifications of the constants in the bound (35)). $\qquad\square$

### A.2 ALMOST SURE CONVERGENCE, NOTATION

To prove convergence, we start with rewriting the processes (13) on a form that is more reminiscent of the integral equations (16). As in Kushner & Yin (2003), we use the convention that

$$\sum_{i=n}^{k} a_i = 0, \ \text{ if } k = n - 1 \text{ (the empty sum)},$$

$$\sum_{i=n}^{k} a_i = - \sum_{i=k+1}^{n-1} a_i, \ \text{ if } k < n - 1.$$

By introducing the function

$$m(t) = \begin{cases} j, \ t_j \le t < t_{j+1}, \\ 0, \ t \le 0, \end{cases} \tag{36}$$

we can write (13) as

$$P_k(t) = p_k + \sum_{i=k}^{m(t_k+t)-1} (p_{i+1} - p_i),$$

$$Q_k(t) = q_k + \sum_{i=k}^{m(t_k+t)-1} (q_{i+1} - q_i). \tag{37}$$

Using the fact that $p_k = P_k(0)$ and $q_k = Q_k(0)$, along with the update (9), we can rewrite (37) as

$$P_k(t) = P_k(0) - \sum_{i=k}^{m(t_k+t)-1} \alpha_i \nabla F(q_i) + M_k(t) - \gamma \sum_{i=k}^{m(t_k+t)-1} \alpha_i \nabla \varphi(p_i),$$

$$Q_k(t) = Q_k(0) + \sum_{i=k}^{m(t_k+t)-1} \alpha_i \nabla \varphi(p_{i+1}), \tag{38}$$

where

$$M_k(t) = \sum_{i=k}^{m(t_k+t)-1} \alpha_i \delta M_i \tag{39}$$

and $\delta M_i = \nabla f(q_i, \xi_i) - \nabla F(q_i)$.

In the next section, we show that the process $\{M_k\}_{k \ge 0}$ converges uniformly on compact sets, almost surely, to $0$.

### A.3 CONVERGENCE OF THE SEQUENCE $\{M_k\}$

The following lemma is and adaptation of part 1 of the proof of Theorem 2.1 from Kushner & Yin (2003):

**Lemma A.3** (Convergence of $\{M_k(t)\}_{k\geq 0}$). *Suppose that Assumption 1, 2 and 6 holds, along with either Assumption 3, 4 or 5. Then, the sequence $\{M_k(t)\}_{k\geq 0}$ converges uniformly on compact sets almost surely to 0. More precisely, for any $T$ it holds that*

$$\lim_{k\to\infty}\sup_{t\in[0,T]}\|M_k(t)\|_2 = 0, \tag{40}$$

*almost surely.*

*Proof of Lemma A.3.* Closely following the proof of Theorem 2.1 in Kushner & Yin (2003): We let $\mathcal{F}_j = \sigma(\xi_1,\ldots,\xi_j)$. By definition, we have that

$$M_k(t) = \sum_{i=k}^{m(t_k+t)-1} \alpha_i \delta M_i,$$

where $\delta M_i = \nabla f(q_i,\xi_i) - \nabla F(q_i)$. Define

$$\tilde{M}_j = \sum_{i=k}^{j} \alpha_i \delta M_i.$$

We will show that $\tilde{M}_j$ is a martingale sequence. We first note that

$$\mathbb{E}\left[\tilde{M}_{j+1}|\mathcal{F}_j\right] = \tilde{M}_j,$$

by Assumption 1.iv) and the fact that the noise is independent. Next, we demonstrate that

$$\mathbb{E}\left[\|\tilde{M}_{j+1}\|_2\right] < \infty, \tag{41}$$

Note that

$$\mathbb{E}\left[\|\tilde{M}_l\|_2^2\right] = \mathbb{E}\left[\|\sum_{i=k}^{l}\alpha_i\delta M_i\|_2^2\right] = \mathbb{E}\left[\sum_{i=k}^{l}\alpha_i^2\|\delta M_i\|_2^2 + 2\sum_{i=k}^{l}\sum_{j=k}^{i-1}\alpha_i\alpha_j\langle\delta M_i,\delta M_j\rangle\right]$$

$$= \mathbb{E}\left[\sum_{i=k}^{l}\alpha_i^2\|\delta M_i\|_2^2\right],$$

where we have used the fact that for $j < i$

$$\mathbb{E}\left[\langle\delta M_i,\delta M_j\rangle\right] = \mathbb{E}\left[\mathbb{E}\left[\langle\delta M_i,\delta M_j\rangle|\mathcal{F}_j\right]\right] = \mathbb{E}\left[\langle\mathbb{E}\left[\delta M_i|\mathcal{F}_j\right],\delta M_j\rangle\right] = 0,$$

since $M_j$ is $\mathcal{F}_j$-measurable and $\mathbb{E}\left[\delta M_i|\mathcal{F}_j\right] = 0$ (recall that $\xi_i$ is independent of $\mathcal{F}_j$). In the case that Assumption 4.ii) holds we therefore have that

$$\mathbb{E}\left[\|\delta M_i\|_2^2\right] < \infty. \tag{42}$$

If instead Assumption 3.ii) holds, we have that

$$\mathbb{E}\left[\|\delta M_{i+1}|\mathcal{F}_i\|_2^2\right] \leq \kappa(F(q_k) - F_*) + (1+\tau)\|\nabla F(q_k)\|_2^2 + \sigma^2.$$

Under Assumption 3.i) or 4.i) we get from Theorem 5.8 that the expectation of the right-hand side is finite[4], in which case (42) also holds. Hence $\tilde{M}_j$ satisfies (41) and it is thus a martingale.

We now show that (40) holds. For any interval $[0,T]$, we have that

$$\sup_{t\in[0,T]}\|M_k(t)\|_2 = \sup_{k\leq j\leq l}\|\tilde{M}_j\|_2,$$

where $l = m(t_k + T)$. By *Doob's submartingale inequality* (Kushner & Yin, 2003; Williams, 1991), we have for every $\mu > 0$ that

$$\mathbb{P}\left(\sup_{k\leq j\leq l}\|\tilde{M}_j\|_2 \geq \mu\right) \leq \frac{\mathbb{E}\left[\|\tilde{M}_l\|_2^2\right]}{\mu^2}.$$

---

[4]Under assumption 3.i) we can use Lemma B.6 to bound the gradient with $2L(F(q_k) - F_*)$ which is bounded in expectation by Theorem 5.8.

which implies that

$$\mathbb{P}\left(\sup_{k \leq j}\|\tilde{M}_j\|_2 \geq \mu\right) \leq C \sum_{i=k}^{\infty} \alpha_i^2$$

and hence

$$\lim_{k \to \infty} \mathbb{P}\left(\sup_{k \leq j}\|\tilde{M}_j\|_2 \geq \mu\right) = 0.$$

By Theorem 1 in Section 2.10.3 of Shiryaev (2016), the sequence $\tilde{M}_j$ converges almost surely to 0, i.e. there is a set $U$ such that $\mathbb{P}(U) = 0$ and for every $\omega \in U^c$ we have that (40) holds. $\qquad\square$

### A.4 Equicontinuity of the sequences $\{P_k\}_{k \geq 0}$ and $\{Q_k\}_{k \geq 0}$

**Lemma 5.10** (Equicontinuous in the extended sense). *Consider $\{Z_k\}_{k \geq 0} = \{(P_k, Q_k)\}_{k \geq 0}$ where the sequences $\{P_k\}_{k \geq 0}$ and $\{Q_k\}_{k \geq 0}$ are defined by (13) (equivalently, by (37)). Suppose that $\{p_k\}_{k \geq 0}$ and $\{q_k\}_{k \geq 0}$ are defined by (9), and that the Hamiltonian is on the form (7). Further, let Assumptions 1, 2 and 6 be valid, as well as either Assumption 3, 4 or 5. Then $\{Z_k\}_{k \geq 0}$ is equicontinuous in the extended sense, almost surely.*

To show Lemma 5.10, we make use of an equivalent definition of extended equicontinuity:

**Lemma A.4** (Equivalent definition of extended continuity). *A sequence of functions $\{f_k\}_{k \geq 0}$, $f_k : \mathbb{R} \to \mathbb{R}^d$ is equicontinuous in the extended sense if and only if $\{|f_k(0)|\}_{k \geq 0}$ is bounded and for every $T$ and $\epsilon > 0$ there is a null sequence $(a_k)_{k \geq 0}$ (that is, $\lim_{k \to \infty} a_k = 0$) such that*

$$\sup_{0<|t-s|\leq\delta,\ t,s\in[0,T]}|f_k(t) - f_k(s)| \leq \epsilon + a_k. \tag{43}$$

*Proof of Lemma A.4.* By definition (14) is equal to

$$\lim_{k \to \infty} b_k \leq \epsilon$$

with

$$b_k := \sup_{j \geq k}\ \sup_{0<|t-s|\leq\delta,\ t,s\in[0,T]}|f_j(t) - f_j(s)|.$$

Define

$$a_k = \max\{0, b_k - \epsilon\}.$$

Then $(a_k)_{k \geq 0}$ satisfies all the requirements; $a_k$ is clearly positive and by continuity of the function $\max\{0, x\}$ it holds that

$$\lim_{k \to \infty} a_k = \max\{0, \lim_{k \to \infty} b_k - \epsilon\} = 0,$$

as $\lim_{k \to \infty} b_k - \epsilon \leq 0$. Furthermore, we have that

$$\sup_{0<|t-s|\leq\delta,\ t,s\in[0,T]}|f_k(t) - f_k(s)| \leq b_k \leq \epsilon + a_k,$$

for every $k$ and thus we have shown that (43) follows from (14). We now show the converse. Suppose that (43) holds. Taking the supremum of (43) we obtain

$$\sup_{j \geq k}\ \sup_{0<|t-s|\leq\delta,t,s\in[0,T]}|f_j(t) - f_j(s)| \leq \epsilon + \sup_{j \geq k} a_j.$$

We finally take the limit with respect to $k$

$$\lim_{k \to \infty}\sup_{j \geq k}\ \sup_{0<|t-s|\leq\delta,\ t,s\in[0,T]}|f_j(t) - f_j(s)| \leq \epsilon + \lim_{k \to \infty}\sup_{j \geq k} a_j =: \epsilon + \limsup_{k \to \infty} a_k.$$

But as $\lim_{k \to \infty} a_k$ exists by assumption and is equal to 0, we have $\limsup_{k \to \infty} a_k = \lim_{k \to \infty} a_k = 0$. We thus conclude that (14) holds. $\qquad\square$

We now turn to the proof of Lemma 5.10.

*Proof of Lemma 5.10.* Closely following Lemma 2 in Freise (2016): We want to show that the sequence $\{Z_k\}_{k \geq 0} = \{(P_k, Q_k)\}_{k \geq 0}$, where $\{P_k\}_{k \geq 0}$ and $\{Q_k\}_{k \geq 0}$ are defined by (13), is equicontinuous in the extended sense.

First, we note that the sequences $\{P_k(0)\}$ and $\{Q_k(0)\}$ are finite except on a set of measure 0, since by Theorem 5.8 $\sup_k \|p_k\|_2 < \infty$ and $\sup_k \|q_k\|_2 < \infty$ almost surely.

By Lemma A.4 an equivalent charaterization of extended equicontinuity is that for every $\epsilon > 0$, there is a sequence $\{a_k\}_{k \geq 0}$ such that $\lim_{k \to \infty} a_k = 0$ and a $\delta > 0$ such that

$$\sup_{|t-s| < \delta, \ t,s \in [0,T]} \|Z_k(t) - Z_k(s)\|_{\ell^2(\mathbb{R}^{2d})} \leq \epsilon + a_k, \text{ a.s.} \tag{44}$$

By (38), we have that

$$\|P_k(t) - P_k(s)\|_2 \leq C(\omega) \sum_{i=m(t_k+s)}^{m(t_k+t)-1} \alpha_i + \|M_k(t)\|_2 + \|M_k(s)\|_2, \tag{45}$$

where $C(\omega) = \sup_i \|\nabla F(q_i) - \gamma \nabla \varphi(p_i)\|_2$. By the boundedness of $p_k$ and $q_k$ along with the continuity of $\nabla F$ and $\nabla \varphi$, we have that $C(\omega) < \infty$, a.s. The sum on the right-hand side of (45) can be rewritten as

$$\sum_{i=m(t_k+s)}^{m(t_k+t)-1} \alpha_i = t_{m(t_k+t)} - t_{m(t_k+s)}.$$

By definition of $m$, (36), we have that

$$t_{m(t_k+t)} \leq t_k + t.$$

Thus

$$t_{m(t_k+t)} - t_{m(t_k+s)} \leq t_k + t - t_{m(t_k+s)}. \tag{46}$$

But we also have

$$t_{m(t_k+s)} \leq t_k + s < t_{m(t_k+s)+1},$$

and hence the right-hand side of (46) can be rewritten and bounded as follows

$$t_k + t - (t_k + s) + (t_k + s) - t_{m(t_k+s)} \leq (t - s) + t_{m(t_k+s)+1} - t_{m(t_k+s)}.$$

Now, $t_{m(t_k+s)+1} - t_{m(t_k+s)} = \alpha_{m(t_k+s)+1}$ and hence we see that

$$\|P_k(t) - P_k(s)\|_2 \leq C(\omega)\left(|t - s| + \alpha_{m(t_k+s)+1}\right) + \|M_k(t)\|_2 + \|M_k(s)\|_2.$$

Let $\epsilon$ be greater than 0. There are now two cases. If $C(\omega) = 0$, (43) clearly holds for any $\delta > 0$. If $C(\omega) \neq 0$, then take $\delta > 0$ so small that $C(\omega)\delta < \epsilon$. We then have

$$\sup_{|t-s| < \delta, \ t,s \in [0,T]} \|P_k(t) - P_k(s)\|_2$$

$$\leq \sup_{|t-s| < \delta, t,s \in [0,T]} \left(C(\omega)(|t - s| + \alpha_{m(t_k+s)+1}) + \|M_k(t)\|_2 + \|M_k(s)\|_2\right)$$

$$< \epsilon + C(\omega)\alpha_{m(t_k)+1} + 2\|M_k(T)\|_2.$$

By Lemma A.3, $\lim_{k \to \infty} \|M_k(T)\|_2 = 0$ a.s. and we see that (43) in Lemma A.4 holds almost surely. A similar argument yields an analogous bound for $Q_k$, and by the equivalence of norms on $\mathbb{R}^{2d}$, we obtain (44). $\qquad\square$

In the next section, we show that the processes $\{P_k\}_{k \geq 0}$ and $\{Q_k\}_{k \geq 0}$ can be written as solutions to the integral equations corresponding to (8), plus terms that converge uniformly on compact sets to 0 as $k$ tends to $\infty$.

### A.5 ASYMPTOTIC SOLUTION

**Lemma 5.12** (Asymptotic solutions). *With the same assumptions and notation as in Lemma 5.10, we can write*

$$P_k(t) = P_k(0) - \int_0^t \nabla F(Q_k(s)) \mathrm{d}s - \gamma \int_0^t \nabla \varphi(P_k(s)) \mathrm{d}s + M_k(t) + \mu_k(t),$$

$$Q_k(t) = Q_k(0) + \int_0^t \nabla \varphi(P_k(s)) \mathrm{d}s + \nu_k(t) + \kappa_k(t),$$
(15)

*where the functions $\{M_k\}_{k\geq 0}$, $\{\mu_k\}_{k\geq 0}$, $\{\nu_k\}_{k\geq 0}$ and $\{\kappa_k\}_{k\geq 0}$ converge to 0 uniformly on compact sets almost surely.*

*Proof of Lemma 5.12.* We start with showing that the sum

$$- \sum_{i=k}^{m(t_k+t)-1} \alpha_i \nabla F(q_i)$$

in Equation (38) can be rewritten as

$$- \int_0^t \nabla F(Q_k(s)) \mathrm{d}s + \mu_{1,k}(t),$$

where $Q_k(t)$ is defined by (13) and $\{\mu_{1,k}\}_{k\geq 0}$ is a sequence of functions that tends to 0 uniformly on compact intervals. Consider

$$I_k := - \int_0^t \nabla F(Q_k(s)) \mathrm{d}s.$$

Then, since $t_k + s$ belongs to a single interval $[t_i, t_{i+1})$,

$$I_k = - \int_0^t \left( \nabla F(q_0) I_{(-\infty,t_0)}(t_k + s) - \sum_{i=0}^{\infty} \nabla F(q_i) I_{[t_i,t_{i+1})}(t_k + s) \right) \mathrm{d}s$$

$$= - \int_0^t \left( \nabla F(q_0) I_{(-\infty,t_0-t_k)}(s) - \sum_{i=0}^{\infty} \nabla F(q_i) I_{[t_i-t_k,t_{i+1}-t_k)}(s) \right) \mathrm{d}s.$$

The term $t_0 - t_k$ is always less than or equal to 0. Hence the first term disappears as we are integrating from 0 to $t$. For $i < k$, we have that $t_{i+1} - t_k \leq 0$. We can therefore start the sum at $i = k$, as earlier terms will not contribute to the integral. Thus,

$$I_k = - \int_0^t \left( \sum_{i=k}^{\infty} \nabla F(q_i) I_{[t_i-t_k,t_{i+1}-t_k)}(s) \right) \mathrm{d}s.$$

Now suppose $t_j - t_k \leq t < t_{j+1} - t_k$. We split up the previous integral as follows:

$$I_k = - \int_0^{t_j-t_k} \left( \sum_{i=k}^{j-1} \nabla F(q_i) I_{[t_i-t_k,t_{i+1}-t_k)}(s) \right) \mathrm{d}s - \int_{t_j-t_k}^t \nabla F(q_k) I_{[t_j-t_k,t_{j+1}-t_k)}(s) \mathrm{d}s$$

$$= - \sum_{i=k}^{j-1} \nabla F(q_i) \alpha_i - \nabla F(q_j) \left( t - t_j + t_k \right),$$

where we have used that $\int_0^{t_j-t_k} I_{[t_i-t_k,t_{i+1}-t_k)}(s) \mathrm{d}s = \alpha_i$. Using the fact that $m(t + t_k) = j$ (where $m(t)$ is defined by (36)), we can rewrite this further as

$$- \sum_{i=k}^{m(t_k+t)-1} \nabla F(q_i) \alpha_i - \mu_{1,k}(t),$$

where $\mu_{1,k}(t) = \nabla F(q_{m(t_k+t)}) \left(t - t_{m(t_k+t)} + t_k\right)$. The function $\mu_{1,k}$ is piecewise linear and $0$ at $t = t_j - t_k$. The gradient $\nabla F$ is Lipschitz-continuous by assumption, and thus there is some positive random variable $C(\omega)$, finite almost everywhere, such that

$$\|\nabla F(q_{m(t_k+t)})\|_2 \leq C(\omega) < \infty,$$

since by Theorem 5.8 $\sup_k \|q_k\|_2 < \infty$. Hence, it holds that

$$\|\mu_{1,k}(t)\|_2 \leq C(\omega)|\alpha_{m(t_k+t)}|.$$

Now, for fixed $T$, we have that $\lim_{k\to\infty} \sup_{t\in[0,T]} \alpha_{m(t_k+t)} = 0$ since $\lim_{k\to\infty} \alpha_k = 0$, and thus $\mu_{1,k}$ converges to $0$ uniformly on compact intervals. Hence, it holds that

$$-\int_0^t \nabla F(Q_k(s))ds = -\sum_{i=k}^{m(t_k+t)-1} \nabla F(q_i)\alpha_i + \mu_{1,k}(t),$$

where

$$\mu_{1,k}(t) = \nabla F(q_{m(t_k+t)})(t_{m(t_k+t)} - t - t_k).$$

In a similar fashion we obtain that

$$-\sum_{i=k}^{m(t_k+t)-1} \alpha_i \nabla\varphi(p_i) = -\int_0^t \nabla\varphi(P_k(s))\mathrm{d}s + \mu_{2,k}(t),$$

where $\{\mu_{2,k}\}_{k\geq 0}$ converges uniformly on compact sets to $0$. Letting $\mu_k = \mu_{1,k} + \gamma\mu_{2,k}$, we obtain the expression in the first line of (15).

We now turn our attention to the second line of (15). By an argument analogous to the previous, we can write the second line of (38) as

$$Q_k(t) = Q_k(0) + \int_0^t \varphi(P_{k+1}(s))\mathrm{d}s + \nu_k(t),$$

where $\nu_k$ converges uniformly on compact sets to $0$. We can rewrite the integral on the right-hand side as

$$\int_0^t \nabla\varphi(P_{k+1}(s))\mathrm{d}s = \underbrace{\int_0^t \nabla\varphi(P_{k+1}(s)) - \nabla\varphi(P_k(s))\mathrm{d}s}_{:=\kappa_k(t)} + \int_0^t \nabla\varphi(P_k(s))\mathrm{d}s.$$

The norm of $\kappa_k(t)$ can be bounded as follows:

$$\|\kappa_k(t)\|_2 \leq \int_0^t \lambda\|P_{k+1}(s)) - P_k(s)\|_2\mathrm{d}s = \int_0^t \lambda\|P_k(\alpha_k + s) - P_k(s)\|_2\mathrm{d}s,$$

where we have used the Lipschitz continuity of $\nabla\varphi$ and the fact that $P_{k+1}(s) = P_k(\alpha_k + s)$. Since $\{P_k(t)\}_{k\geq 0}$ is equicontinuous in the extended sense by Lemma 5.10, there is for each $T$ and $\epsilon > 0$ a $\delta > 0$ such that

$$\limsup_{k\to\infty} \sup_{|t-s|<\delta,\ t,s\in[0,T]} \|P_k(\alpha_k + s) - P_k(s)\|_2 \leq \epsilon. \tag{47}$$

What remains is to show is that $\{\kappa_k\}_{k\geq 0}$ converges uniformly on compact sets to $0$. For any $T$, we have that

$$\lim_{k\to\infty} \sup_{t\in[0,T]} \|\kappa_k(t)\|_2 \leq \lim_{k\to\infty} \int_0^T \lambda\|P_k(\alpha_{k+1} + s) - P_k(s)\|_2\mathrm{d}s$$

since the integrand is positive. By Theorem 5.8, we can bound $\|P_k(\alpha_{k+1} + s) - P_k(s)\|_2 \leq 2\sup_{t\in\mathbb{R}}\|P_k(t)\|_2 < \infty$. Thus, we can use the Lebesgue dominated convergence theorem and take the limit inside the integral:

$$\lim_{k\to\infty} \sup_{t\in[0,T]} \|\kappa_k(t)\|_2 \leq \int_0^T \lim_{k\to\infty} \lambda\|P_k(\alpha_{k+1} + s) - P_k(s)\|_2\mathrm{d}s$$

By (47), we can make the integrand arbirarily small by choosing $k$ so large that $\alpha_{k+1} \leq \delta$. $\qquad\square$

A.6   CONVERGENCE TO A LOCALLY ASYMPTOTICALLY STABLE SET

The goal of this section is to show

**Theorem 5.16.** *Under the same assumptions and notation as in Theorem 5.6, let $A$ be a locally asymptotically stable set for (8). If there exists a compact set in the domain of attraction of $A$ that $\{z_k\}_{k\geq 0}$ visits infinitely often, then $z_k \to A$ almost surely:*

$$\lim_{k\to\infty} \inf_{a\in A} \|z_k - a\|_{\ell^2(\mathbb{R}^{2d})} = 0, \ \ a.s. \tag{17}$$

We start with showing the following help-lemma:

**Lemma A.6.** *In the same context as Theorem 5.16, for each $\delta > 0$ there is a subsequence $\{z_{n_k}\}_{k\geq 0}$ of $\{z_k\}_{k\geq 0}$ such that that $\{z_{n_k}\}_{k\geq 0} \subset N_\delta(A)$.*

*Proof of Lemma A.6.* Since $\{z_{n_k}\}_{k\geq 0} \subset K$, and $K$ is compact, we can find a further subsequence $\{z_{n'_k}\}_{k\geq 0}$ that tends to $z_0 \in K$. Let $\{Z_{n'_k}\}_{k\geq 0}$ be the sequence of shifted interpolations associated with $\{z_{n'_k}\}_{k\geq 0}$. This family is equicontinuous in the extended sense by Lemma 5.10, and thus it has a subsequence $\{Z_{n''_k}\}_{k\geq 0}$ converging to a function $z(\cdot)$ which is a solution to (8) by Lemma 5.12, and satisfies $z(0) = z_0$. Since $z_0$ is in the domain of attraction of $A$, it holds that

$$\lim_{t\to\infty} \inf_{a\in A} \|z(t) - a\|_{\ell^2(\mathbb{R}^{2d})} = 0.$$

Choose $T_{\frac{\delta}{2}}$ so large that

$$\inf_{a\in A} \|z(t) - a\|_{\ell^2(\mathbb{R}^{2d})} < \frac{\delta}{2} \tag{48}$$

for all $t \geq T_{\frac{\delta}{2}}$. Then, we have

$$\inf_{a\in A} \|Z_{n''_k}(t) - a\|_{\ell^2(\mathbb{R}^{2d})} \leq \|Z_{n''_k}(t) - z(t)\|_{\ell^2(\mathbb{R}^{2d})} + \inf_{a\in A} \|z(t) - a\|_{\ell^2(\mathbb{R}^{2d})}$$

$$< \|Z_{n''_k}(t) - z(t)\|_{\ell^2(\mathbb{R}^{2d})} + \frac{\delta}{2}.$$

Since $\{Z_{n''_k}\}$ converges uniformly on compact sets to $z$, we can choose $N_{\frac{\delta}{2}}$ so large that for any $n'_k \geq N_{\frac{\delta}{2}}$ we have that $\sup_{s\in[0,t]}\|Z_{n''_k}(s) - z(s)\|_{\ell^2(\mathbb{R}^{2d})} < \frac{\delta}{2}$. Hence, for $n''_k > N_{\frac{\delta}{2}}$ we have

$$\inf_{a\in A} \|Z_{n''_k}(t) - a\|_{\ell^2(\mathbb{R}^{2d})} < \delta,$$

which yields the statement of the lemma. □

With the help of Lemma A.6, we now show Theorem 5.16. The proof is inspired by the proof strategy in Fort & Pagès (1996).

*Proof of Theorem 5.16.* Let $\epsilon > 0$ and let $\delta > 0$ be as in Definition 5.15. According to Lemma A.6, there is a subsequence $\{z_{r_k}\}_{k\geq 0}$ of $\{z_k\}_{k\geq 0}$ such that $\{z_{r_k}\}_{k\geq 0} \subset N_{\delta/2}(A)$.

We now show by contradiction that $\{z_k\}_{k\geq 0}$ cannot escape $N_\epsilon(A)$ infinitely often. Suppose that there is a subsequence $\{z_{s_k}\}_{k\geq 0} \subset N_\epsilon(A)^c$.

Define $\ell_0 = \min\{j : z_j \in N_{\delta/2}(A)\}$ and recursively for $k = 1, 2, \ldots,$

$$n_k = \min\{j : j \geq \ell_{k-1} \text{ and } z_j \in N_\epsilon(A)^c\},$$
$$m_k = \max\{j : j \leq n_k \text{ and } z_j \in N_{\delta/2}(A)\},$$
$$\ell_k = \min\{j : j \geq n_k \text{ and } z_j \in N_{\delta/2}(A)\}.$$

Then there is no index $j \in \{m_k + 1, \ldots, n_k\}$ such that $z_j \in N_{\delta/2}(A)$; i.e. $m_k$ is the last index for which $z_j$ visits $N_{\delta/2}(A)$ before going to $N_\epsilon(A)^c$.

Consider the associated sequence of functions $\{Z_{m_k}\}_{k\geq 0}$. This satisfies

$$Z_{m_k}(t) = z_{m_k} \in N_{\delta/2}(A), \qquad\qquad 0 \leq t < \alpha_{m_k},$$
$$Z_{m_k}(t) = z_{n_k} \in N_\epsilon(A)^c, \qquad\qquad t_{n_k} - t_{m_k} \leq t < t_{n_k} - t_{m_k} + \alpha_{n_k}.$$

In between these two time intervals, $Z_{m_k}$ attains the values $z_{m_k+1}, z_{m_k+2}, \ldots, z_{n_k-1}$. We can therefore guarantee that

$$Z_{m_k}(t) \in N_{\delta/2}(A)^c \text{ for } t \in [\alpha_{m_k}, t_{n_k} - t_{m_k}]. \tag{49}$$

Let $\{Z_{m'_k}\}$ be a subsequence of $\{Z_{m_k}\}$ that converges uniformly on compact sets to a function $z$.

First, assume that $\limsup_{k \to \infty} t_{n'_k} - t_{m'_k} = \infty$. Then we can extract a subsequence (which we continue to denote $\{t_{n_k}\}$) for which $\lim_{k \to \infty} t_{n'_k} - t_{m'_k} = \infty$. Under this assumption, it holds that

$$z(t) \in N_{\delta/2}(A)^c \text{ for } t > 0.$$

If this was not the case there would be a $t' > 0$ such that $z(t') \in N_{\delta/2}(A)$. By the openness of $N_{\delta/2}(A)$ we can choose $\eta > 0$ such that $B(z(t'), \eta) \subset N_{\delta/2}(A)$. There is a $K_\eta$ such that $k \geq K_\eta$ implies that

$$\|Z_{m'_k}(t') - z(t')\| < \eta,$$

i.e. $Z_{m'_k}(t') \in B(z(t'), \eta) \subset N_{\delta/2}(A)$. However, since $t_{n'_k} - t_{m'_k} \to \infty$ and $\alpha_{m'_k} \to 0$, it holds that $t' \in [\alpha_{m'_k}, t_{n'_k} - t_{m'_k}]$ for large enough $k$. This contradicts (49), so that indeed $z(t) \in N_{\delta/2}(A)^c$ for $t > 0$. By Theorem 5.11, $z$ is continuous and since $Z_{m_k}(0) \in N_{\delta/2}(A)$ we must thus have $z(0) \in \partial N_{\delta/2}(A)$. However, the fact that $z(t) \in N_{\delta/2}(A)^c$ for $t \geq 0$ contradicts the asymptotic stability of $A$, since this path which starts in $\partial N_{\delta/2}(A) \subset N_\delta(A)$ does not approach $A$. This is a contradiction towards our assumption that $t_{n'_k} - t_{m'_k} \to \infty$, and we can thus define $\tilde{T} = \sup_k t_{n'_k} - t_{m'_k} < \infty$. Then $[0, \tilde{T}]$ is a compact interval such that $\{t_{n'_k} - t_{m'_k}\}_{k \geq 0} \subset [0, \tilde{T}]$. Hence there is a subsequence $\{t_{n''_k} - t_{m''_k}\}_{k \geq 0} \subset \{t_{n'_k} - t_{m'_k}\}_{k \geq 0}$ that converges to some $T \in [0, \tilde{T}]$.

The corresponding sequence of functions $\{Z_{m''_k}\}$ is a subsequence of $\{Z_{m'_k}\}$ and thus it must also converge uniformly on compact sets to the same function $z$. From the uniform convergence it also follows that

$$z_{n''_k} = Z_{m''_k}(t_{n''_k} - t_{m''_k}) \to z(T).$$

Since each $z_{n''_k} \in N_\epsilon(A)^c$ we must have $z(T) \in N_\epsilon(A)^c$. However, this contradicts the Lyapunov stability of $A$ and therefore also our original assumption that there exists a subsequence $\{z_{s_k}\}_{k \geq 0} \subset N_\epsilon(A)^c$. This concludes the proof. $\qquad\square$

### A.7 CONVERGENCE TO A STATIONARY POINT

We will now apply Theorem 5.16 and show that $\{z_k\}_{k \geq 0}$ converges to the set $\{z : H(z) \leq \liminf_k H(z_k)\}$. First, we need to show that it is locally asymptotically stable:

**Lemma 5.17.** *Consider the same assumptions and notation as in Theorem 5.6. For each $c$, if the set $\{z : H(z) \leq c\}$ is non-empty, it is a locally asymptotically stable set for the solutions to (8).*

*Proof.* We need to show that for all $\epsilon > 0$ we can choose $\delta > 0$ so that if $z_0 \in N_\delta(A)$, $z(t)$ stays in $N_\epsilon(A)$ and that $\lim_{t \to \infty} z(t) \in A$.

By Lemma B.3, there exists a $\eta > 0$ such that $\{z : H(z) \leq c + \eta\} \subset N_\epsilon(A)$. Now $z(t)$ will stay in $\{z : H(z) \leq c + \eta\}$ since $z(t)$ decreases along the paths of $H$. However, it might not converge to $A$ since there may exists stationary points $z_*$ such that

$$c < H(z_*) \leq c + \eta,$$

and if we reach one of these we will get stuck there instead of reaching $A$. Define

$$c_* = \inf\{H(z_*) : c < H(z_*) \leq c + \eta, \nabla H(z_*) = 0\}.$$

It holds that $c_* > c$, i.e. we cannot find stationary points for which $H(z_*)$ is arbitrarily close to $c_*$. We can see this by letting $\Lambda = \{x : \nabla H(x) = 0\}$ and $K = [c, c + \eta]$. Then by Assumption 1.iii), there exist numbers $\{y_i\}_{i=1}^n$, such that $y_1 < \cdots < y_n$ and

$$\{H(z) : c \leq H(z) \leq c + \eta : z \in \Lambda\} = H(\Lambda) \cap K = \{y_1, \ldots, y_n\}.$$

If $y_1 = c$, we have that

$$y_2 = \min H(\Lambda) \cap K = \min\{H(z) : c < H(z) \le c + \eta, \; z \in \Lambda\} = c_*$$

and thus $c_* > c$. Similarly, if $y_1 > c$, we also get that $c_* = y_1 > c$. Thus indeed it always holds that $c_* > c$, and we can take $\mu > 0$ so small that $c_* - \mu > c$. By Lemma B.2 there exists some $\delta > 0$ such that

$$N_\delta(A) \subset \{H(z) < c_* - \mu\}$$

Since $H$ is decreasing along the paths of $z(t)$, any solution starting in $N_\delta(A)$ will stay inside $\{z : H(z) \le c_* - \mu\}$ (and thus $N_\epsilon(A)$). By La Salle's invariance principle, any path starting in the compact set $\mathcal{M} = \{z : H(z) \le c_* - \mu\}$ tends to $\{z \in \mathcal{M} : \nabla H(z) = 0\}$. All points $z_* \in \{z \in \mathcal{M} : \nabla F(z) = 0\}$ satisfy $H(z_*) \le c$, by the choice of $c_*$ and $\mu > 0$. Thus, $z(t) \to \{z : H(z) \le c\}$ whenever $z(0) \in N_\delta(A)$.

We have for the given $\epsilon > 0$ found a $\delta > 0$ such that any path in $N_\delta(A)$ never leaves $N_\epsilon(A)$ and tends to $A$ as $t \to \infty$. $\qquad\square$

We are now ready to prove our main result, Theorem 5.6:

**Theorem 5.6.** *Let Assumptions 1, 2 and 6 be satisfied, as well as either Assumption 3, 4 or 5. Then $\{q_k\}_{k \ge 0}$ converges almost surely to the set of stationary points of the objective function $F$. If we additionally assume that Assumption 7 holds, the convergence is to a unique stationary point.*

*Proof.* Let $c = \liminf_k H(z_k)$. We start with showing that

$$\lim_{k \to \infty} H(z_k) = c. \tag{50}$$

By Lemma 5.17, the set $A = \{z : H(z) \le c\}$ is a locally asymptotically stable set, and by Lemma B.5, we can find a compact set $K$ in the domain of attraction of $A$ that $\{z_k\}_{k \ge 0}$ enters infinitely often: In particular, we can take $K = \mathcal{M}$, where $\mathcal{M}$ is as in the proof of Lemma 5.17; $\mathcal{M}$ is in the domain of attraction of $A$, and by Lemma B.5, $\{z_k\}_{k \ge 0}$ visits $\mathcal{M}$ infinitely often. Theorem 5.16 then implies that $z_k \to \{z : H(z) \le c\}$. Suppose that $\lim_k H(z_k) \ne c$. The negation of the statement is

$$\exists \epsilon > 0 : \forall n \exists n_k \ge n, H(z_{n_k}) \le c - \epsilon \vee H(z_{n_k}) \ge c + \epsilon.$$

In the case that there exists a subsequence $\{z_{n_k}\}$ such that $H(z_{n_k}) \ge c + \epsilon$, since then $\{z_k\}$ would not converge to $\{z : H(z) \le c\}$. If there exists a subsequence that satisfies $H(z_{n_k}) \le c - \epsilon$ we would have $\liminf_k H(z_k) \le c - \epsilon < c$ which is also a contradiction by the choice of $c$.

We now know that $\lim_{k \to \infty} H(z_k) = c$, but we have yet to verify that $\{z_k\}_{k \ge 0}$ converges to the set of stationary points. Suppose that this is not the case. For brevity let $\Lambda = \{z : \nabla H(x) = 0\}$. Then there exists an $\epsilon_0 > 0$ and subsequence $\{z_{n_k}\}_{k \ge 0}$ such that

$$\inf_{x \in \Lambda} \|z_{n_k} - x\| \ge \epsilon_0. \tag{51}$$

From the previous paragraph, it holds that

$$\lim_{n_k \to \infty} H(z_{n_k}) = c.$$

By Theorem 5.8, the sequence $\{z_{n_k}\}_{k \ge 0}$ is bounded. By Lemma B.1 we can thus find a further subsequence (still denoted by $\{z_{n_k}\}_{k \ge 0}$) and a point $\tilde{z}_0$ such that $\lim_{n_k \to \infty} z_{n_k} = \tilde{z}_0$ and $H(\tilde{z}_0) = c$. The sequence of interpolations $\{Z_{n_k}(\cdot)\}$ associated with $\{z_{n_k}\}$, has a subsequence $\{Z_{n'_k}(\cdot)\}$ that converges to a solution $\tilde{z}(\cdot)$, such that $\tilde{z}(0) = \tilde{z}_0$. By (51) it holds that $\tilde{z}_0 \notin \{z : \nabla H(z) = 0\}$. As $H$ is decreasing along the paths of $\tilde{z}(\cdot)$, we have for $t' > 0$ that $c = H(\tilde{z}_0) = H(\tilde{z}(0)) > H(\tilde{z}(t'))$. However, $\tilde{z}(\cdot)$ is taking values in $L(\{z_k\})$, the set of limit points of $\{z_k\}$, compare Proposition 1.b) in Fort & Pagès (1996). Thus, there is some subsequence $\{z_{m_k}\}$ that converges to $H(\tilde{z}(t'))$. But since $c = H(\tilde{z}_0) > H(\tilde{z}(t'))$ and $\{z_{m_k}\}$ converges to $H(\tilde{z}(t'))$ which is a contradiction, by the choice of $c$. It follows that the set of limit points $L(\{z_k\}_{k \ge 0})$ of $\{z_k\}_{k \ge 0}$ is contained in $\{z : \nabla H(z) = 0\}$. Since $z_{k+1} - z_k \to 0$, the limit set $L(\{z_k\}_{k \ge 0})$ is connected (Asic & Adamovic, 1970). By Assumption 7, this implies that $\{z_k\}_{k \ge 0}$ converges to a single stationary point. $\qquad\square$

At last, we prove Corollary 5.7:

**Corollary 5.7** (Convergence in expectation). *Let Assumptions 1, 2, 6 and 7 be valid. Further, let the Hamiltonian be on the form (7) and let the sequences $\{p_k\}_{k\geq0}$ and $\{q_k\}_{k\geq0}$ be generated by (9). Then it holds under that*

$$\lim_{k\to\infty} \mathbb{E}\left[\|\nabla F(q_k)\|_2^\theta\right] = 0,$$

*where $\theta = 1$ under Assumption 3 and $\theta = \frac{1}{2}$ under Assumption 4 or 5.*

*Proof.* By Theorem 5.6 $\{q_k\}_{k\geq0}$ converges almost surely to a random variable $q_*$ which takes values in the set of stationary points of $F$. Hence,

$$\lim_{k\to\infty} \|\nabla F(q_k)\|_2 = 0, \text{a.s.},$$

compare Lemma 2.3 in van der Vaart (2000). From Lemma B.6 we have that

$$\|\nabla F(q_k)\|_2^2 \leq 2L\left(F(q_k) - F_*\right).$$

By Theorem 5.8, we obtain that

$$\sup_k \mathbb{E}\left[\|\nabla F(q_k)\|_2^2\right] < \infty.$$

By Lemma 3 in Chapter 2.6 of Shiryaev (2016) we obtain (taking $G(t) = t^2$) that the sequence $\{\|\nabla F(q_k)\|\}_{k\geq0}$ is uniformly integrable. It follows from Theorem 5 in Chapter 2.6 of Shiryaev (2016) that

$$\lim_{k\to\infty} \mathbb{E}\left[\|\nabla F(q_k)\|_2\right] = 0.$$

Under Assumption 4 or 5, we instead get from Lemma B.6 and Theorem 5.8 that

$$\sup_k \mathbb{E}\left[\|\nabla F(q_k)\|_2\right] < \infty.$$

It follows that

$$\lim_{k\to\infty} \mathbb{E}\left[\|\nabla F(q_k)\|_2^{\frac{1}{2}}\right] = 0.$$

$\square$

# B AUXILIARY RESULTS

Several of the results in this section are relatively standard, but we keep them here for the sake of reference.

**Lemma B.1.** *Let $\{x_k\}_{k\geq0}$ be a sequence in $\mathbb{R}^d$. Suppose that $\sup_k\|x_k\| < \infty$ and that $f(x_k) \to y$, where $f : \mathbb{R}^d \to \mathbb{R}$ is continuous. Then there is a subsequence $\{x_{n_k}\}_{k\geq0} \subset \{x_k\}_{k\geq0}$ that converges to some number $x$ such that $f(x) = y$*

*Proof.* Since $\sup_k\|x_k\| < \infty$ there is some compact set $K$ such that $\{x_k\} \subset K$. By compactness, there is some subsequence $\{x_{n_k}\}$, that converges to some element $x$. The sequence $\{f(x_{n_k})\}$ is a subsequence of $\{f(x_k)\}$, and must converge to the same limit $y$. However, by continuity of $f$, we have that $\lim_{k\to\infty} f(x_{n_k}) = f(\lim_{k\to\infty} x_{n_k}) = f(x)$. Thus, $f(x) = y$. $\square$

The next two Lemmas, Lemma B.2 and B.2, are helpful in showing that the sublevel sets the Hamiltonian are locally asymptotically stable:

**Lemma B.2.** *Suppose that $f : \mathbb{R}^d \to \mathbb{R}$ is continuous and coercive. Let $A = \{x : f(x) \leq c\}$, where $c$ is such that $A \neq \emptyset$. Then, for every $\eta > 0$ there is $\delta > 0$ such that $N_\delta(A) \subset \{x : f < c + \eta\}$.*

*Proof.* We first note that since $f$ is coercive, $A$ is compact. Let $\eta > 0$ be given. By continuity of $f$, there is $\delta > 0$ such that

$$|x - y| < \delta \implies |f(x) - f(y)| < \eta.$$

For such $\delta$, we consider

$$N_\delta(A) = \{x : \inf_{a \in A} \|x - a\| < \delta\}.$$

Take $x_0 \in N_\delta(A)$. Then $\inf_{a \in A} \|x_0 - a\| < \delta$ and by the definition of the infimum, there exists for each $n$ and element $a_n \in A$ such that

$$\|x_0 - a_n\| < \inf_{a \in A} \|x_0 - a\| + \frac{1}{n}.$$

Then $\{a_n\} \subset A$ and by compactness there is a subsequence $\{a_{n_k}\}$ that converges to an element $a_* \in A$. Since

$$\|x_0 - a_{n_k}\| < \inf_{a \in A} \|x_0 - a\| + \frac{1}{n_k},$$

it holds that

$$\|x_0 - a_*\| \leq \inf_{a \in A} \|x_0 - a\| < \delta.$$

Since $f$ is continuous, we have that

$$f(x_0) < f(a_*) + \eta \leq c + \eta$$

i.e. $x_0 \in \{x : f(x) \leq c + \eta\}$. Thus $N_\delta(A) \subset \{x : f(x) \leq c + \eta\}$. $\qquad\square$

**Lemma B.3.** *Let* $A = \{x : f(x) \leq c\}$ *(where $c$ is such that $A \neq \emptyset$). Then for every $\epsilon > 0$ there is $\eta > 0$ such that* $\{x : f(x) \leq c + \eta\} \subset N_\epsilon(A)$.

*Proof.* If this was not the case, then there exists some $\epsilon > 0$ and for every $n$ we can find $x_n$ that satisfies

$$x_n \in \{x : f(x) \leq c + \frac{1}{n}\} \cap N_\epsilon(A)^c \subset \{x : f(x) \leq c + 1\} \cap N_\epsilon(A)^c.$$

The latter is compact since $N_\epsilon(A)^c$ is closed and $\{x : f(x) \leq c + 1\}$ is compact. Thus, $\{x_n\}$ has a subsequence $\{x_{n_k}\}$ that converges to $x_* \in \{x : f(x) \leq c + 1\} \cap N_\epsilon(A)^c \subset N_\epsilon(A)^c \subset A^c$. However, each $x_{n_k} \in \{x : f(x) \leq c + \frac{1}{n_k}\}$ and thus by continuity it holds that $f(x_*) \leq c$ which is a contradiction. $\qquad\square$

We will use the next Lemma to show that under the given assumptions, the sublevel set $\{z : H(z) \leq \liminf_{k \to \infty} H(z_k)\}$ is non-empty:

**Lemma B.4.** *Let* $f : \mathbb{R}^d \to \mathbb{R}$ *be a continuous function that is bounded below by* $f_* = \inf_{x \in \mathbb{R}^d} f(x)$. *Let* $\{x_k\}_{k \geq 0}$ *be a sequence in* $\mathbb{R}^d$ *such that* $\sup_k \|x_k\| < \infty$. *Put* $c = \liminf_k f(x_k)$. *Then* $\{x : f(x) \leq c\} \neq \emptyset$.

*Proof.* By assumption $\{x_k\}_{k \geq 0}$ is contained in some compact set $K$. By continuity, it holds that $C = \sup_k f(x_k) < \infty$ and hence the sequence $\{f(x_k)\}_{k \geq 0}$ is contained the compact set $[f_*, C]$. It follows that $c \in [f_*, C]$. By a standard result in real analysis, we can (since $\{f(x_k)\}_{k \geq 0}$ is bounded) find a subsequence $\{f(x_{n_k})\}$ that converges to $c$. By Lemma B.1, there exists a further subsequence $\{x_{n'_k}\}$ that converges to some element $x$ such that $f(x) = c$. Hence $x \in \{x : f(x) \leq c\}$ and $\{x : f(x) \leq c\} \neq \emptyset$. $\qquad\square$

**Lemma B.5.** *Let $f$ be a function which is bounded below. Put* $c = \liminf_k f(x_k)$. *Then for every $\delta > 0$, the sequence $\{x_k\}$ is in the set* $A_\delta = \{x : f(x) \leq c + \delta\}$ *infinitely often.*

*Proof.* The negation of statement is

$$\neg \left( \forall \delta > 0, \forall n, \exists n_k, n_k \geq n \land z_{n_k} \in A_\delta \right)$$

which can be rewritten as

$$\exists \delta_0 > 0, \exists k_0, \forall k \geq k_0, \ x_k \notin A_\delta.$$

This means that for all $k \geq k_0$,

$$f(x_k) > c + \delta_0. \tag{52}$$

Taking the infimum, we see that

$$\inf_{k \geq k_0} f(x_k) \geq c + \delta_0. \tag{53}$$

Since $\inf_{k \geq k_0} f(x_k)$ is increasing, we have for $k \geq k_0$ that

$$\inf_{m \geq k} f(x_m) \geq \inf_{m \geq k_0} f(x_m) \geq c + \delta_0$$

Taking the supremum over $k$, we see that we must have

$$\liminf_k f(x_k) = \sup_{k \geq 0} \inf_{m \geq k} f(x_m) \geq c + \delta_0,$$

i.e. $\liminf_{k \to \infty} f(x_k) \geq c + \delta_0$, which is a contradiction. $\qquad\square$

**Lemma B.6.** *Let $F$ be bounded from below by $F_*$. If $F$ is $(L_0, L_1)-$smooth, it holds that*

$$\|\nabla F(q)\| \leq 2L_1(F(q) - F_*) + \frac{L_0}{L_1}.$$

*If $F$ is instead $L$-smooth with Lipschitz constant $L$, it holds that*

$$\|\nabla F(q)\|_2^2 \leq 2L(F(q) - F_*).$$

*Proof.* Consider first the $(L_0, L_1)-$smooth case. Put

$$q_+ = q - \frac{1}{L_1 \|\nabla F(q)\|} \nabla F(q), \tag{54}$$

Then

$$\|q_+ - q\| = \frac{1}{L_1}$$

Thus, the conditions for (21) in Zhang et al. (2020a) are satisfied, and it holds that

$$F(q_+) - F(q) \leq \langle \nabla F(q), q_+ - q \rangle + \frac{L_0 + L_1 \|\nabla F(q)\|}{2} \|q_+ - q\|^2.$$

Inserting (54) into the previous expression we see that

$$F(q_+) - F(q) \leq -\frac{1}{L_1} \|\nabla F(q)\| + \frac{L_0 + L_1 \|\nabla F(q)\|}{2} \frac{1}{L_1^2},$$

Rearranging the terms, we find that

$$F(q_+) - F(q) \leq -\frac{1}{2L_1} \|\nabla F(q)\| + \frac{L_0}{2} \frac{1}{L_1^2}$$

One more rearrangement yields

$$\|\nabla F(q)\| \leq 2L_1(F(q) - F(q_+)) + \frac{L_0}{L_1}.$$

Since $F(q_+) \geq F_*$ we obtain the statement of the first part of the Lemma. The proof of the second part is very similar but simpler, and therefore omitted. $\qquad\square$

**Lemma B.7.** *Let $F$ satisfy Assumption 5.i) and $f(\cdot, \xi)$ satisfy Assumption 5.iii). Then*

$$\|\nabla f(x,\xi)\|_2 \leq 2NL_1\left(F(q) - F_*\right) + \frac{L_0}{L_1}, \tag{55}$$

*almost surely, where $F_* = \frac{1}{N}\sum_{i=1}^N \inf_{q\in\mathbb{R}^d} f_i(x)$.*

*Proof.* We start with showing that there exists a constant $C$ such that

$$f(x,\xi) - \inf_{x\in\mathbb{R}^d} f(x,\xi) \leq C(F(x) - F_*), \tag{56}$$

almost surely. First we note that by the properties of $\inf$ it holds that

$$\inf_{x\in\mathbb{R}^d} f(x,\xi) = \inf_{x\in\mathbb{R}^d}\left(\frac{1}{|B_\xi|}\sum_{i\in B_\xi} f_i(x)\right) = \frac{1}{|B_\xi|}\cdot\inf_{x\in\mathbb{R}^d}\left(\sum_{i\in B_\xi} f_i(x)\right) \geq \frac{1}{|B_\xi|}\cdot\sum_{i\in B_\xi}\inf_{x\in\mathbb{R}^d} f_i(x).$$

Hence

$$f(x,\xi) - \inf_{x\in\mathbb{R}^d} f(x,\xi) = \frac{1}{|B_\xi|}\cdot\sum_{i\in B_\xi} f_i(x) - \inf_{x\in\mathbb{R}^d} f(x,\xi)$$

$$\leq \frac{1}{|B_\xi|}\cdot\sum_{i\in B_\xi} f_i(x) - \frac{1}{|B_\xi|}\cdot\sum_{i\in B_\xi}\inf_{x\in\mathbb{R}^d} f_i(x)$$

Since $f_i(x) - \inf_{x\in\mathbb{R}^d} f_i(x)) \geq 0$, the previous expression can be bounded by

$$\frac{1}{|B_\xi|}\cdot\sum_{i=1}^N f_i(x) - \inf_{x\in\mathbb{R}^d} f_i(x) = \frac{N}{|B_\xi|}\left(F(x) - F_*\right).$$

As the batch size $|B_\xi|$ is non-decreasing, (56) holds. Since $f(\cdot, \xi)$ is $(L_0, L_1)-$smooth, it holds that

$$\|\nabla f(x,\xi)\|_2 \leq 2L_1\left(f(x,\xi) - \inf_{x\in\mathbb{R}^d} f(x,\xi)\right) + \frac{L_0}{L_1}.$$

Combining the previous expression with (56), we obtain (55). $\qquad\square$

*Remark B.8.* Note that for fixed, *deterministic* $x$, (55) implies that the norm $\|\nabla f(x,\xi) - \nabla F(x)\|_2$ is bounded almost surely. This means that around a stationary point $q_*$ or for the initial iterate $q_0$ (which in this paper is assumed to be deterministic), the norm of the noise is not heavy-tailed. When $x = q_k$ is a random variable this is no longer the case. This is in line with e.g (Gurbuzbalaban et al., 2021), in which it is reported that the noise is not heavy-tailed initially.

## C NUMERICAL EXPERIMENTS

In order to illustrate the behavior of the algorithms, we set up three numerical experiments. The experiments are implemented in Tensorflow 2.12 (Abadi et al., 2015). We consider the following kinetic energy functions

- $\varphi(x) = x$. (Abbreviated as SHD.)

- $\varphi(x) = \sqrt{\epsilon + \|x\|_2^2}$. (Normalized).
- $\varphi(x) = \sqrt{1 + \|x\|_2^2}$. (SoftClipped.)

In the plots we also see the results of Adam (Kingma & Ba, 2015), SGD with momentum (abbreviated as SGDmom), Clipped SGD with momentum (ClippedSGDmom) and Clipped SGD (ClippedSGD). All of these algorithms are as implemented in Abadi et al. (2015). Each of the experiments were run for 4 random seeds ranging from 3000 to 3003, that all yielded similar results. In the plots we see the results for the random seed 3000. For every experiment we consider a grid of initial step sizes $\beta$ with values

$$\left(10^{-4}, 5\cdot 10^{-4}, 10^{-3}, 5\cdot 10^{-3}, 10^{-2}, 5\cdot 10^{-2}, 0.1, 0.5, 1\right).$$

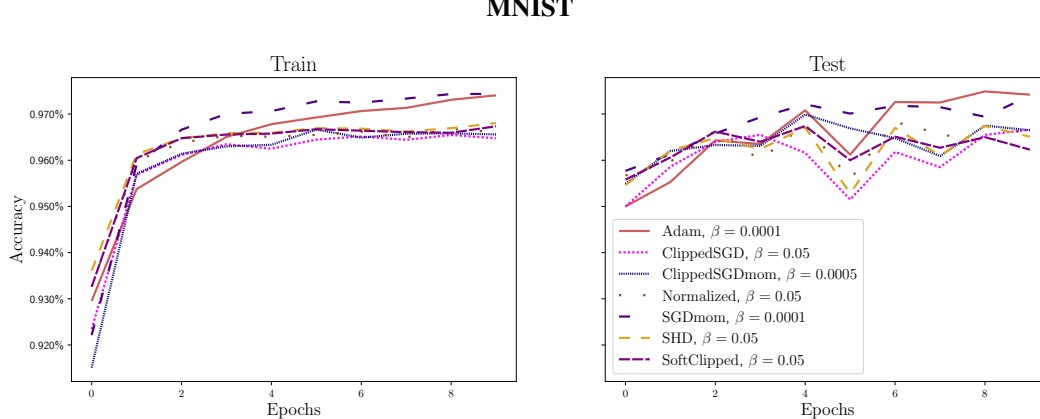

Figure 1: Accuracy of the different methods when used for training a simple convolutional neural network to classify the MNIST dataset. Each method displays the result when the optimal initial step size $\beta$ is used.

To find the optimal initial value of $\beta$ among these, we use the Keras implementation of the Hyperband algorithm (Li et al., 2018); a hyper parameter optimization algorithm that makes use of a combination of random search and successive halving (Jamieson & Talwalkar, 2016). In all the experiments, we use a step size scheme defined by $\frac{\beta}{\lfloor k/10 \rfloor + 1}$, where $\beta$ is the initial step size and $k$ is the epoch.

## C.1 CLASSIFICATION OF THE MNIST DATASET

The first experiment is a simple convolutional neural network used to classify the MNIST dataset (Lecun et al., 1998). We split the data in the standard way, but use both the training and validation sets for training. The training- and test accuracy after 20 epochs is displayed in Figure 1. All of the algorithms work well for the given problem. Around the 10th epoch several of the methods see an improvement in training accuracy due to the step size decrease. All the methods converge relatively fast on both training and test data and display performance on par to the state of the art algorithms implemented in Tensorflow. We also remark that Normalized and SoftClipped perform at their best with a higher step size, like the clipped SGD-methods. The methods all exhibit a smooth behavior on the training data, while the oscillations are slightly higher on the test data.

### C.1.1 DETAILS ON THE NETWORK ARCHITECTURE

The model consists of one convolutional layer with 32 filters, a kernel size of 3 and a stride of 1. Padding is chosen such that the input has the same shape as the output. Upon this, a dense layer of 128 neurons is stacked before the output layer with a softmax function. The activation function used in the hidden layers is the *exponential linear unit* (Clevert et al., 2016). In both the convolutional- and the dense layers we use a weight decay of $5 \cdot 10^{-3}$.

## C.2 CLASSIFICATION OF THE CIFAR10 DATASET

The second experiment is a VGG-network (Simonyan & Zisserman, 2015) used to classify the CIFAR10 dataset (Krizhevsky, 2009). We split the data in the standard way, but use both the training and validation sets for training. In Figure 2, we see the train- and test accuracy for the methods. We see that all the kinetic energy functions display performance on par with state of the art algorithms. On the training data, the majority of the methods converge to a stationary point for which the models has an accuracy of about 70 percent. After the first step size decrease, the algorithms find a new stationary point towards which they converge. The training curves are smooth, while again the oscillations are slightly higher on the test data during the first 15 epochs. Adam, Normalized and SoftClipped exhibits a smoother behavior on the test data than the other algorithms.

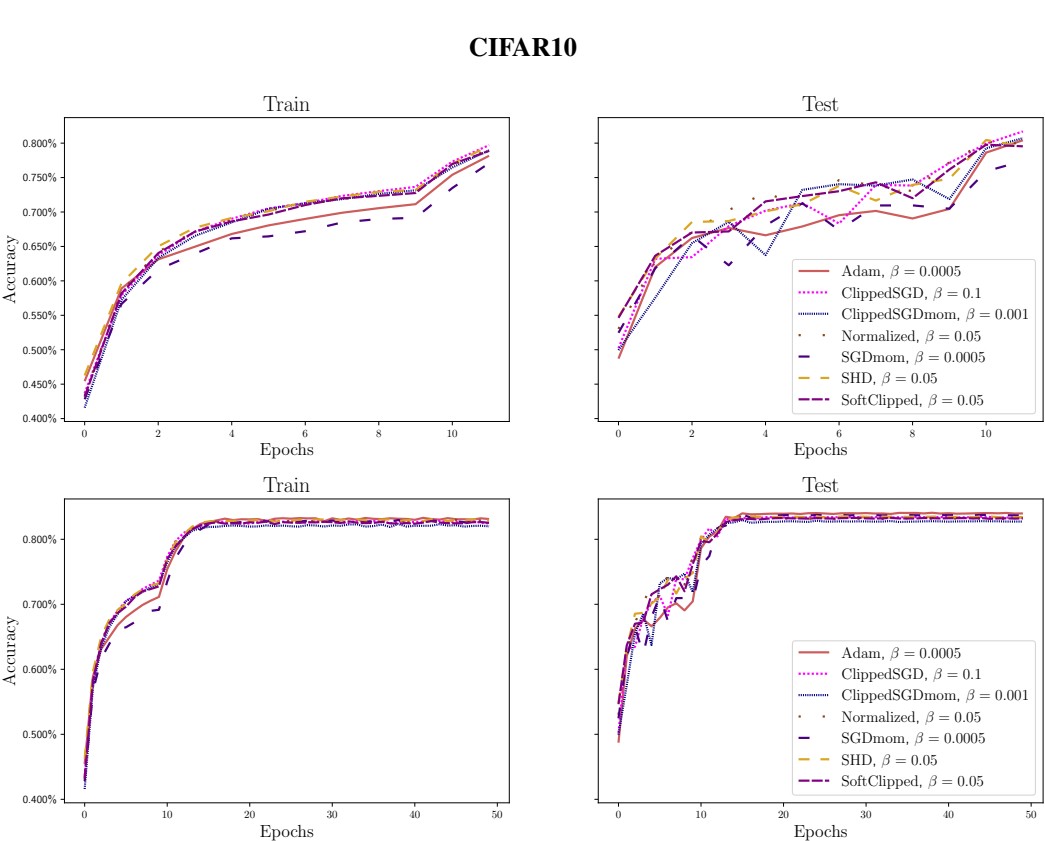

Figure 2: Accuracy of the different methods when used for training a VGG-network to classify the CIFAR10 dataset. The two plots in the top shows the first 12 epochs and the two plots in the bottom, all the 50 epochs. Each method displays the result when the optimal initial step size $\beta$ is used.

**Penn. Treebank**

Figure 3: Perplexity of the different methods when used for text prediction on the Penn. Treebank dataset. Each method displays the result when the optimal initial step size $\beta$ is used.

### C.2.1 DETAILS ON THE NETWORK ARCHITECTURE

The model consists of three blocks of convolutional layers. The first block consists of two convolutional layers with 32 filters with kernel size of 3, each followed by a batch normalization layer (Ioffe & Szegedy, 2015). This is then passed through a max-pooling layer with a kernel size of $2 \times 2$ and a stride of 2. In the convolutional layers a weight decay of $5 \cdot 10^{-3}$ is used. The next two blocks have similar structure but with filter sizes of 64 and 128 respectively. In between each layer a drop out of 20% is used. As in the first example we use a dense hidden layer with 128 neurons before the output layer. In all layers, the exponential linear unit was used as activation function.

### C.3 TEXT PREDICTION ON THE PENNSYLVANIA TREEBANK CORPUS

The last experiment is a long-short-term memory-type model, that we use for text prediction on the Pennsylvania Treebank portion of the Wall Street Journal corpus (Marcus et al., 1993). The design of the experiment is inspired by similar ones in e.g. Graves (2014); Mikolov et al. (2012); Pascanu et al. (2012); Zhang et al. (2020a). For the experiment, we use the same training and validation split of the dataset as in Merity et al. (2018).[5]

In Figure 3 we see the exponentiated average regret, or *perplexity*

$$\exp\left(\frac{1}{K}\sum_{k=1}^{K} f(q_k, \xi_k)\right),$$

where K is the number of batches in an epoch. For a model that chooses each of the words in the vocabulary with uniform probability we expect this to be close to the size of the vocabulary (in this case 10000). We expect a well performing model to have a perplexity close to 1. In Figure 3, we see the training- and test perplexity for the various methods. The SHD-method achieves a slightly higher perplexity on the training data then the other methods. (Although this behavior is not as pronounced on the test data). In general, methods that make use of some sort of normalization or clipping appears to be working best for this task; the best method is the SoftClipped, which quickly reaches the lowest perplexity on the test data set.

### C.3.1 DETAILS ON THE NETWORK ARCHITECTURE

The network consists of an embedding layer of size 400 upon which three bidirectional LSTM-layers are stacked, each with 1150 RNN-units. A dropout of 50% is used in the LSTM-layers, as well as

---

[5]We call the validation set 'Test' in Figure 3 so that it agrees with the terminology in the previous experiments.

weight decay of $1.2 \cdot 10^{-6}$. In the output layer, a dense layer with $10000$ neurons is used. The batch size is $64$ and we use a sequence length of $10$ words.

## C.4 CONCLUSIONS

The experiments in the previous section verify the theoretical results in the paper and we see that most of the algorithms also exhibit performance on par with state of the art algorithms. We remark that in all the examples, we used very generic networks for the sake of finding problems on which we could easily compare the behavior of the models. Better performance could be achieved in all cases if the networks and optimizers would have been tuned more carefully to the classification problems, but the intention here is to illustrate the behavior of the algorithms rather than achieving state of the art results.

