# OpenReview forum: "Almost sure convergence of stochastic Hamiltonian descent methods"
_ICLR.cc/2025/Conference — Submitted to ICLR 2025_

### Official Review · Reviewer_AXMi · 2024-10-27

**Soundness:** 3
**Presentation:** 3
**Contribution:** 2
**Rating:** 6
**Confidence:** 4

**Summary:**

This paper shows the almost sure convergence of a class of stochastic Hamiltonian descent methods, including gradient normalization and soft clipping algorithms, under three different set of assumptions which provide validity of the conclusions to a wide range of settings. The proof relies on the typical argument followed in dynamic systems and control theory, which is to: (1) find a suitable Lyapunov function (in this case the Hamiltonian of the system); and (2) invoke LaSalle's Invariance theorem to establish the convergence to a (isolated) stationary point the (non-convex) objective function.

Typo: Line 112 should read "term" instead of "tern"

**Strengths:**

The paper is well written, structured, and thus easy to follow. It contributes to the theoretical understanding of SGD with momentum under gradient normalization and soft clipping.

Although I didn't read every proof in extreme detail, I believe the formal arguments and results are correct.

Minor detail: Since the authors initially present the formulation of the nearly Hamiltonian system in continuous time (see eq. (8)), perhaps the choice of the Lyapunov function in the proof of Theorem 5.7 could be motivated in that context, showing that: $\dot V = -\gamma\||\nabla\varphi\||^2 \leq 0.$

**Weaknesses:**

The main issue I have with the paper is that, because the proof strategy closely follows the approach developed by Kushner and Yin (2003) (cf. Section 5 and Theorem 5.2.1), it is not immediately clear why their proof cannot be directly invoked after showing that the sequence of iterates is finite almost surely. For example, Lemma A.3 can in fact be found in the first part of Theorem 5.2.1 but this is not explicitly mentioned by the authors. Could the authors explicitly discuss how their proof extends or differs from Kushner and Yin (2003), for example by including a paragraph comparing their approach to Kushner and Yin's and highlighting any novel elements or modifications needed for the Hamiltonian specific setting.

**Questions:**

* Could the authors more clearly explain how their approach seems to extend Kushner and Yin (2003)? This would allow the reader to better evaluate the novel elements needed in the proof almost sure convergence.
* This next question is not related to the perceived weakness of the paper but more for a deeper understanding of the results. Because the kinetic energy function is assumed to be differentiable, almost sure convergence cannot be concluded for clipping as studied by Zhang et. al. (2020) in expectation. Could the authors explain the need of the differentiability assumption, and why working with the sub-gradients of $\varphi$ is not possible?

---

> ### Author Response · Authors · 2024-11-15
>
> We thank the reviewer for providing valuable feedback.
>
> Regarding the choice of Lyapunov function in Theorem 5.7:
>
> The reviewer is right in that this is a natural choice since it is also a Lyapunov function for the system (8). We will state this more clearly in a revised version of the article.
>
> Regarding what sets the contribution apart from Kushner & Yin (2003):
>
> We cannot invoke Theorem 5.1 of Kushner & Yin (2003) directly as the algorithm is an implicit-explicit discretization of an ODE. The solution is to show that the ”difference” between the implicit and explicit evaluation of the algorihtm
> \begin{align*}
> \kappa_k(t) = \int_0^t \nabla \varphi(P_{k+1}(s)) ds -\int_0^t \nabla \varphi(P_{k}(s)) ds
> \end{align*}
> converges to $0$ uniformly on compact intervals. This is (part of) the proof of Theorem 5.11. We will try to make this clearer in the article.
> Other reasons that  Theorem 5.1 cannot be applied directly is that the noise assumption (i.e. A2.1 in section 5 of Kushner & Yin (2003)) would have to be verified in Setting 1 and 3.
> The assumption that the iterates are almost surely bounded is also very strong (and something we show holds for the algorithms in the paper).
>
>
> Regarding clipping as studied in Zhang et al. (2020):
>
> It is likely possible to analyse this kind of clipping as well, but then one probably would have to work with differential inclusions rather than ODE:s. We agree that this is an interesting perspective and something to consider for future studies. We consider differentiable functions since this is what people tend to implement in practice. For instance the experiments in Zhang et al. (2020) are implemented using a soft-clipping function.

---

> ### Comment · Reviewer_AXMi · 2024-11-26
>
> I thank the authors for their comments and clarifications. I maintain my score.

---

### Official Review · Reviewer_KB2A · 2024-11-03

**Soundness:** 3
**Presentation:** 4
**Contribution:** 2
**Rating:** 3
**Confidence:** 5

**Summary:**

The paper investigates the almost sure convergence of stochastic Hamiltonian descent methods in optimization, especially in the context of machine learning and statistical estimation. Key methods discussed include gradient normalization and soft clipping, which are used to stabilize stochastic gradient descent (SGD) with momentum, commonly applied in non-convex optimization settings. The authors analyze the convergence properties of these modified SGD algorithms, especially when applied to objective functions with heavy-tailed noise and potentially infinite gradient variance, by classifying these techniques into dissipative Hamiltonian systems.

 One of the main contributions of the paper is to show that gradient normalization and soft clipping can be viewed as stochastic implicit-explicit Euler discretizations of dissipative Hamiltonian systems. By utilizing Hamiltonian dynamics and dynamical systems theory, the authors provide a unified convergence framework for these methods in different objective function settings, including \(L\)-smooth and \((L_0, L_1)\)-smooth functions.

The paper introduces assumptions for the objective function and the stochastic gradient, such as coercivity and locally bounded variance, which ensure that the optimization problem satisfies the necessary conditions for convergence. Moreover, the convergence guarantee is extended to settings with heavy-tailed noise, which is common in real data. In particular, the analysis shows that the iterative updates of these modified SGD algorithms converge almost surely to the set of stationary points of the objective function under these assumptions.

The proof strategy involves the use of a Lyapunov function based on the Hamiltonian of the system to demonstrate the finiteness and boundedness of the iterative sequences. The analysis then uses an ODE (Ordinary Differential Equation) approach to show that these sequences converge almost surely to stationary points of the objective function, and not just in expectation. This result is important because it ensures convergence along each individual optimization path and not just on average, which is crucial for the robustness of optimization algorithms.

**Strengths:**

- The paper is very well-written, and the authors rigorously verify the standard assumptions of the ODE approach, even if this means including assumptions that may seem idealized.

**Weaknesses:**

The paper doesn’t bring substantial new insights to the field; it essentially revisits classical methods, line by line, and demonstrates the applicability of standard ODE-based techniques. While it is certainly rigorous and methodical, the paper ultimately falls short of deepening our understanding of the algorithms themselves or providing fresh perspectives on their behavior. This makes it a rather conventional contribution, adhering closely to established approaches without pushing beyond them in terms of theoretical or practical insight.

The authors rely on the Kushner and Yin approach, which, although mathematically elegant, has a significant drawback: it depends on assumptions that are notably difficult to verify. For example, Theorem 5.15 includes the assumption that “there exists a compact set in the domain of attraction of \( A \) that \( \{z_k\}_{k \geq 0} \) visits infinitely often.” However, the practicality of checking this condition is questionable. For dissipative systems with Lyapunov functions, more robust and accessible conditions are typically available. For instance, Benaïm’s 2006 work on the dynamics of stochastic approximation provides an alternative perspective with less restrictive assumptions, and the work by Andrieu, Moulines, and Priouret (2005) presents stability criteria that are generally easier to validate. Both of these references suggest that a more flexible framework is possible and may have been preferable in analyzing the convergence behavior.



See Benaïm, M. (2006). Dynamics of stochastic approximation algorithms. In Seminaire de probabilites XXXIII (pp. 1-68). Berlin, Heidelberg: Springer Berlin Heidelberg.
- Andrieu, C., Moulines, É., & Priouret, P. (2005). Stability of stochastic approximation under verifiable conditions. SIAM Journal on control and optimization, 44(1), 283-312. "

**Questions:**

- How do you check in pratice Assumption 1-iii). Give a clear and easy to check criterion (based a.g. on Sard's theorem)
- How do you check the conditions of Theorem 5.15

---

> ### Author Response · Authors · 2024-11-15
>
> We thank the reviewer for coming with constructive criticism as well as raising some valid points of concern.
>
> Regarding the contribution of the paper:
>
> The reviewer is right in that the proof closely follows that of Kushner & Yin (2003) and some parts are included for the sake of completeness and the conveniece of the reader since Kushner & Yin (2003) is sparse with details. This if for instance the case with Theorem 5.15.
>
> The results of Kushner & Yin (2003) or Benaïm (2006) does not work right away since the schemes we consider are implicit-explicit discretizations of an ODE while Kushner & Yin (2003)
> considers explicit discretizations. This is Lemma 5.11 and 5.9 in which we show that the shifted sequence of interpolations are equi-continuous in the extended sense (even though the discretization is implicit-explicit).
>
> Another part of the novelty of the paper lies in that we consider objective functions that are $(L_0,L_1)-$smooth and pose less restrictive assumptions on the noise, adapted to the machine learning setting. To the best of our knowledge $(L_0,L_1)-$smooth functions are not considered in previous works that make use of the ODE method. We show that the iterates are bounded a.s. in these settings and that it is possible to apply the extension of the ODE method.
>
> Regarding the verification of the assumptions of Theorem 5.15:
>
> The reviewer is correct in that it is in general a strong assumption that the iterates enter a certain compact set in the domain of attraction of the locally asymptotically stable set of the theorem.
> We do show in the proof of Theorem 5.5 that this is the case for the algorithms in the paper (under the given assumptions) and that we can apply Theorem 5.15.
>
> Regarding Assumption 1.iii):
>
> The question that the reviewer raises is valid; it is difficult to verify this assumption in practice
> and we could have elaborated more on this in the article. We will add a remark in the revised version of the manuscript.
> It is a technical assumption that rules out pathological behaviour; it is slightly stronger than that of e.g. Proposition 6.4 in Benaïm (2006) but weaker than that of Prop. 3.2 in Benaïm (1996).
> In many cases the function has isolated equilibria and in this case Assumption 1.iii) is satisfied. That the equilibria are isolated is necessary to obtain convergence to a unique stationary point.
>
> Regarding using the approach of Benaïm (2006):
>
> It is true that this approach can be used as well and it may be the case some assumptions are less restrictive. (Such as those discussed in the previous paragraph). We prefer the approach of Kushner & Yin (2003)  as less ”technical machinery” is needed, e.g. the notions of asymptotic pseudo-trajectories and chain recurrence. We also think that this approach is more intuitive (extracting convergent subsequences of $\{Z_k\}_{k \geq0}$ and appealing to the Arzela—Ascoli theorem etc.) This is entirely our opinion of course.
>
>
> Benaïm, M. (1996). A dynamical systems approach to stochastic approximation.
> Benaïm, M. (2006). Dynamics of stochastic approximation algorithms.
> Kushner, H.J. & Yin, G. (2003) Stochastic approximation and recursive algorithms and applications.

---

> > ### Comment · Reviewer_KB2A · 2024-11-26
> >
> > I find the book by Kushner and Yin (2003) extremely hard to digest— - it contains notable errors, especially in the use of differential inclusions in the context of projections, but that is another topic. Their approach essentially relies on earlier, almost certain convergence results by Kushner-Clarke.
> >
> > In my opinion, Benaïm's notion of pseudotrajectories is much more intuitive and practical. If you study Benaïm's proofs, you come across familiar tools such as the Arzelà-Ascoli theorem and relative compactness arguments. The dynamics of asymptotic pseudotrajectories, in particular the structure of internally chain recurrent sets, allows the recovery of Kushner-Clarke results without having to resort to "unusual" assumptions (see Section 6 of Benaïm). For example, Proposition 6.4 in Benaïm (2006) provides a much more "reasonable" characterization of the convergence points when a Lyapunov function is available. This condition is also verifiable, e.g. by applying Sard's theorem in the context of gradient algorithms. In contrast, I have always struggled to imagine how to rigorously prove the Kushner-Clarke condition.
> >
> > You mention that "the function has isolated equilibria in many cases"," but even this is not so easy to prove in general!
> >
> > That said, I see novelty in your approach, particularly in the way you handle the implicit-explicit discretization. While the adaptation itself is relatively straightforward, the rigor with which the paper is written stands out and is something I really appreciate.
> > I simply find the scope of the paper limited, with results that are difficult to apply due to the presence of assumptions that are impossible to verify. Its relevance to the machine learning community, in my opinion, is limited. However, with some reworking, it could make for a solid paper in a mathematics journal, especially if a more modern perspective on stochastic approximation is adopted.

---

> > > ### Author Response · Authors · 2024-11-27
> > >
> > > We very much appreciate the reviewer's positive comments about the novelty and rigour!
> > >
> > > We realize that we probably won't change the reviewer's opinion on this, but we still think the manuscript would be of interest to the machine learning community, and this is why we submitted it to ICLR. The ML area is typically where clipping schemes of this form are encountered, and in particular the convergence results for (L_0,L_1)-smooth functions with heavy tailed noise are aimed at cost functionals typically appearing in ML applications.
> > >
> > > We do acknowledge that it might be hard to verify that the equilibria are isolated for a general optimization problem, but for ML problems there are techniques for handling this. For example, it does hold if the Hessian is non-degenerate at the equilibrium, and it is argued in e.g. [1,2,3] that introducing regularization, batch-normalization, or skip connections in neural networks
> > > may lead to such well-behaved problems. This is of course by no means a guarantee that it is satisfied for all ML problems. However, there are many applications of interest for which it is reasonable to expect it to be fulfilled.
> > >
> > > With this being said, we do agree on Kushner & Yin being sparse on details. That is why we decided to provide more detailed and rigorous proofs, for the convenience of the readers.
> > >
> > > [1] Jia, Z. & Su, H. (2020). Information-Theoretic Local Minima Characterization and Regularization
> > >
> > > [2] Orhan, A. E. & Pitkow, X. (2018). Skip connections eliminate singularities.
> > >
> > > [3] Mehta et al. (2018). The Loss Surface Of Deep Linear Networks Viewed
> > > Through The Algebraic Geometry Lens.

---

### Official Review · Reviewer_1tRZ · 2024-11-04

**Soundness:** 4
**Presentation:** 4
**Contribution:** 3
**Rating:** 6
**Confidence:** 2

**Summary:**

The paper studies a family of stochastic gradient methods given by equation 9 and for this family almost sure convergence to stationary points is proved under the following settings:
1. Smooth objective functions with stochastic gradients having locally bounded variance,
2. $(L_0,L_1)$-smooth objective functions with stochastic gradients having finite second moments,
3. $(L_0,L_1)$-smooth objective functions occurring in the empirical risk minimization problem with stochastic gradients having bounded expectation.

Two interesting algorithms that are part of this family are SGD with soft clipped momentum and SGD with normalized momentum.

**Strengths:**

The paper is written in an easy to understand way and the ideas flow smoothly between sections. On the technical side it has the following strengths:
1. Previous work has given guarantees that hold either in expectation or with high probability. These do not guarantee the convergence of every trajectory. However, the results presented in this paper can guarantee the convergence of almost all the trajectories i.e. there exists a set of initial points of measure zero whose trajectories are not guaranteed to converge.
2. The analysis done in previous work holds true only under strong assumption that the stochastic gradients are bounded. Whereas the current work’s analysis holds under much more general assumption of bounded variance.

**Weaknesses:**

The paper focuses primarily on proving almost sure convergence and does not provide any claims about convergence rate. The paper could benefit from showing convergence rate of the family of methods given by equation 9 under one of the settings.

**Questions:**

1. (Clarification for Assumption 4 iii) This assumption is not satisfied when $\varphi(x)=||x||^2/2$. For the family of algorithms under study defined by equation 9 to include SGD with momentum we need $\varphi(x)= ||x||^2/2$. So, the analysis does not hold true for SGD with momentum. Is this correct?
2. (Suggestion about numerical experiments in Appendix C) The main claim of the paper that the family of algorithms converges to a stationary point of the objective function can be better demonstrated if there are graphs showing the evolution of $||\nabla F||_2$ with the number of epochs.

---

> ### Author Response · Authors · 2024-11-15
>
> We would like to thank the reviewer for their kind and constructive feedback on our article. The reviewer correctly points out that we do not present any results regarding convergence rates. We agree that this is also interesting and something we consider for future work. The current focus of the article is to investigate the almost sure convergence of clipping algorithms and the interplay with Hamiltonian dynamics. To the best of our knowledge, statements about the order of convergence for stochastic algorithms applied to non-convex functions are typically local in nature. Specifically, such results hold within a neighborhood of a stationary point or for a subsequence of iterates (e.g., $\min_{1 \leq k \leq K} \lVert \nabla F(q_k)\rVert_2 \in \mathcal{O}(\dots)$). In contrast, the results presented in our article are global in the sense that they apply regardless of the location of the iterates and hold for the entire sequence ${q_k}_{k \geq 0}$.
>
> Regarding the ”clipping assumption” (Assumption 4.iii):
> This assumption is made in Setting 2 and 3 when the objective function is assumed to be $(L_0,L_1)-$smooth.  When the objective function is merely L-smooth (in Setting 1) this assumption is not needed. Hence SGD with momentum is covered by the analysis of Setting 1 but not that of Setting 3 and 4.  We believe that this illustrates the fact that clipping algorithms are a better option for ”(L_0,L_1)-”smooth functions. We will add a remark about this in a revised version of the manuscript.
> Regarding numerical experiments:
> We’re presenting the experiments in a similar form to what seems to be most common in the field. It is essentially equivalent to reporting the gradient as the reviewer suggests. (We see that we get convergence in the plot). If the reviewer has very strong opinions about adding numerical experiments that show the evolution of $\lVert \nabla F\rVert$ we would of course be open to consider this. We are however not sure that this would contribute significantly to the overall value of the paper and would merit the invested time and computational effort.

---

> ### Comment · Reviewer_1tRZ · 2024-11-25
>
> I thank the authors for addressing my concerns.
>
> The paper sets out to prove almost sure convergence of stochastic Hamiltonian descent methods and does so. They also extend the Kushner & Yin (2003) proof strategy to the case of implicit explicit discretization of ODE.

---

### Author Response · Authors · 2024-11-25
**Revised manuscript**

We have now uploaded a revised version of the manuscript, in which we have added the four remarks which we promised in our answers to the referees. They are contained in Remark 5.2, Remark 5.5, the third paragraph of Section 5.1, and the paragraph directly after Theorem A.1.

---

> ### Author Response · Authors · 2024-11-28
>
> Just so there is no confusion: we did a last-minute update of the PDF in order to not exceed the 10-page limit. There was no change of content, just very minor reformulations of two sentences in Section 4.1.

---

### Meta-Review · Area_Chair_rFv5 · 2024-12-22

**Metareview:**

This paper considers gradient normalization and soft clipping methods in the context of Hamiltonian systems. Authors view these as a kind of Euler discretization of dissipative systems and use dynamical systems approach and show convergence to stationary points of the objective function, under several interesting regimes.


This paper was reviewed by three expert reviewers and received the following Scores/Confidence: 6/2, 6/4, 3/5. I think paper is studying an interesting topic but authors are not able to convince the reviewers sufficiently well about the novelty of their work. The following concerns were brought up by the reviewers:

- The main concern about this paper was its novelty in terms of insights it brings to the field.
- Authors rely on classical techniques in this area, which in turn depend on conditions that are difficult to verify or too opaque in practice.


Although two reviewers were positive, no reviewers championed the paper and they are not particularly excited about the paper. As such, based on the reviewers' suggestion, as well as my own assessment of the paper, I recommend not including this paper to the ICLR 2025 program.

**Additional Comments On Reviewer Discussion:**

Authors provided a response which was ultimately deemed insufficient by the main critical reviewer.

---

### Decision · Program_Chairs · 2025-01-22

Reject